# Biosynthesis and Significance of Fatty Acids, Glycerophospholipids, and Triacylglycerol in the Processes of Glioblastoma Tumorigenesis

**DOI:** 10.3390/cancers15072183

**Published:** 2023-04-06

**Authors:** Jan Korbecki, Mateusz Bosiacki, Izabela Gutowska, Dariusz Chlubek, Irena Baranowska-Bosiacka

**Affiliations:** 1Department of Biochemistry and Medical Chemistry, Pomeranian Medical University in Szczecin, Powstańców Wlkp. 72, 70-111 Szczecin, Poland; jan.korbecki@onet.eu (J.K.); mateusz.bosiacki@pum.edu.pl (M.B.); izabela.gutowska@pum.edu.pl (I.G.); dchlubek@pum.edu.pl (D.C.); 2Department of Anatomy and Histology, Collegium Medicum, University of Zielona Góra, Zyty 28 Str., 65-046 Zielona Góra, Poland; 3Department of Functional Diagnostics and Physical Medicine, Faculty of Health Sciences, Pomeranian Medical University in Szczecin, Żołnierska 54 Str., 71-210 Szczecin, Poland

**Keywords:** glioblastoma, brain tumor, fatty acid, polyunsaturated fatty acid, lipid droplets, triacylglycerol, docosahexaenoic acid, glycerophospholipids

## Abstract

**Simple Summary:**

This review discusses the synthesis and significance of fatty acids, glycerophospholipids, and triacylglycerol in glioblastoma. The focus is on all enzymes involved in the synthesis of these lipids, highlighting their roles in the tumorigenesis of glioblastoma. Due to the fact that the role of many of these enzymes in glioblastoma tumorigenesis remains unexplored, we conducted a bioinformatic analysis based on the GEPIA database to indicate their possible functions and significance. Specific properties of certain enzymes were also described to indicate their functions in the tumorigenesis of glioblastoma better.

**Abstract:**

One area of glioblastoma research is the metabolism of tumor cells and detecting differences between tumor and healthy brain tissue metabolism. Here, we review differences in fatty acid metabolism, with a particular focus on the biosynthesis of saturated fatty acids (SFA), monounsaturated fatty acids (MUFA), and polyunsaturated fatty acids (PUFA) by fatty acid synthase (FASN), elongases, and desaturases. We also describe the significance of individual fatty acids in glioblastoma tumorigenesis, as well as the importance of glycerophospholipid and triacylglycerol synthesis in this process. Specifically, we show the significance and function of various isoforms of glycerol-3-phosphate acyltransferases (GPAT), 1-acylglycerol-3-phosphate O-acyltransferases (AGPAT), lipins, as well as enzymes involved in the synthesis of phosphatidylcholine (PC), phosphatidylethanolamine (PE), phosphatidylserine (PS), phosphatidylinositol (PI), and cardiolipin (CL). This review also highlights the involvement of diacylglycerol O-acyltransferase (DGAT) in triacylglycerol biosynthesis. Due to significant gaps in knowledge, the GEPIA database was utilized to demonstrate the significance of individual enzymes in glioblastoma tumorigenesis. Finally, we also describe the significance of lipid droplets in glioblastoma and the impact of fatty acid synthesis, particularly docosahexaenoic acid (DHA), on cell membrane fluidity and signal transduction from the epidermal growth factor receptor (EGFR).

## 1. Introduction

Glioblastoma belongs to the highest IV grade of the central nervous system (CNS) tumors in the current World Health Organization (WHO) classification [1]. The incidence of this cancer is about 4 cases per 100,000 population per year [2,3,4]. It has a very poor prognosis, with a median survival of less than a year for patients with this cancer [4,5,6]. The treatment of glioblastoma involves surgical removal of the tumor, radiotherapy, and chemotherapy, most commonly using temozolomide (TMZ) [7] as well as lomustine and carmustine [8]. Antiangiogenic therapy, including bevacizumab, an antivascular endothelial growth factor (VEGF) antibody, is also used [9,10]. In recent years, tumor-treating fields (TTFields) have been introduced into glioblastoma treatment [8,11]. The technique involves the use of alternating electric fields that disrupt the proliferation of glioblastoma cells during mitosis.

The prognosis for patients with glioblastoma has shown improvement with the introduction of new therapeutic approaches [8]. However, glioblastoma remains one of the most aggressive and challenging types of cancer, with very poor outcomes. In light of this, researchers are investigating the molecular mechanisms of glioblastoma to deepen our understanding of tumorigenesis in glioblastoma tumors and to advance current therapies or develop novel ones.

One promising direction of research focuses on the metabolic activity of cancer cells in glioblastoma, with a particular interest in fatty acid synthesis [12,13,14]. Fatty acids are essential structural elements of glioblastoma cancer cells and can act as lipid mediators. Despite its significance, a comprehensive review of the role of fatty acid synthesis in the tumorigenesis of glioblastoma is currently lacking.

This article aims to synthesize and summarize the current knowledge on the importance of fatty acid synthesis in the development and progression of glioblastoma.

## 2. Synthesis of Fatty Acids and Glioblastoma

### 2.1. Synthesis of Fatty Acids

In human cells, de novo fatty acid synthesis begins with fatty acid synthase (FASN) [15]. Another important enzyme involved in de novo fatty acid synthesis is acetyl-CoA carboxylase (ACC), which catalyzes the carboxylation of acetyl-Coenzyme A (CoA) using bicarbonate (HCO_3_^−^) and ATP [16]. The product of ACC enzymatic activity is malonyl-CoA, which is utilized by FASN to elongate the synthesized acyl-CoA by two carbon units. Due to the fact that palmitoyl-CoA C16:0 cannot be further elongated in FASN, other enzymes are involved in the biosynthesis of longer acyl-CoAs. Acyl-CoA is elongated by a two-carbon unit in four consecutive enzymatic reactions: condensation, reduction, dehydration, and reduction [17]. These reactions are catalyzed respectively by the elongation of long-chain fatty acid family members 1-7 (Elovl1-7) [17], 3-ketoacyl-CoA reductase (KAR) [18], 3-hydroxyacyl-CoA dehydratase [19], and trans-2,3-enoyl-CoA reductase (TER) [18]. Condensation is the rate-limiting step of elongation. This reaction is catalyzed by elongases and requires, similarly to FASN, malonyl-CoA [20,21]. In humans, seven different elongases are distinguished, with enzymes in this group differing in terms of their preferred substrate [22].

Another group of enzymes involved in the synthesis of fatty acids is desaturases, which catalyze the formation of double bonds between carbons in the acyl-CoA hydrocarbon chain. Desaturases can be divided into two subgroups based on their substrate preference: saturated fatty acyl-CoA desaturases and polyunsaturated fatty acyl-CoA desaturases [17]. The former includes stearoyl-CoA desaturase (SCD), of which there are two isoforms in humans and four in mice [17]. These enzymes participate in the synthesis of monounsaturated fatty acids (MUFA) from saturated fatty acids (SFA) and are located in the endoplasmic reticulum [23,24]. They exhibit Δ^9^-desaturase activity, which creates a double bond between carbons 9 and 10, counting from the carboxyl group. SCD shows activity for saturated fatty acyl-CoAs with carbon chain lengths between 12 and 18, with the greatest activity for palmitoyl-CoA C16:0 and stearoyl-CoA C18:0 [25]. Humans, but not rodents, also have another isoform of the enzyme, stearoyl-CoA desaturase 5 (SCD5) [23]. Like SCD, this enzyme is located in the endoplasmic reticulum [23] and has the same substrate specificity [26,27,28], leading to the synthesis of palmitoleoyl-CoA C16:1n-7 and oleoyl-CoA C18:1n-9 from the appropriate saturated fatty acyl-CoA. The second group of desaturases is the polyunsaturated fatty acyl-CoA desaturases [29], which are involved in the synthesis of polyunsaturated fatty acids (PUFA) [17]. The PUFA biosynthesis pathway includes two desaturases: fatty acid desaturase (FADS)1/Δ^5^-desaturase (D5D) and FADS2/Δ^6^-desaturase (D6D) [30,31,32,33].

FADS1 and FADS2, together with Elovl2 and Elovl5, participate in PUFA synthesis. However, the n-3 and n-6 series of PUFA are not synthesized de novo but are produced by elongation and desaturation of the hydrocarbon chain of other polyunsaturated fatty acyl-CoA of the same series from the diet (Figure 1). Thus, arachidonic acid (ARA) C20:4n-6 is synthesized from linoleoyl-CoA C18:2n-6 or γ-linolenoyl-CoA C18:3n-6. Similarly, eicosapentaenoic acid (EPA) C20:5n-3 and docosahexaenoic acid (DHA) C22:6n-3 are synthesized from α-linolenoyl-CoA C18:3n-3 [34].

FADS3 is also present in humans, but its role is unclear. It does not cause the desaturation of saturated fatty acyl-CoA and therefore does not participate in MUFA production [35]. It also does not directly cause the desaturation of polyunsaturated fatty acyl-CoA. FADS3 may play a role in regulating the production of PUFA [36], although its activity is still debated. Previous studies have shown that FADS3 may cause Δ^13^-desaturation of *trans*-vaccenic acid [35,37], although other studies have not confirmed this [36]. FADS3 may also have Δ^14^Z-desaturase activity towards 1-deoxysphinganine [38].

### 2.2. Significance of the Fatty Acid Synthesis Enzymes in Glioblastoma

#### 2.2.1. Fatty Acid Synthase and Acetyl-CoA Carboxylase in Glioblastoma

In glioblastoma cancer cells, glucose and glutamine are the sources of carbon for fatty acid production [39,40]. Glucose is converted into pyruvate through glycolysis and then into acetyl-CoA via the pyruvate dehydrogenase complex (PDC). Glutamine is transformed into α-ketoglutarate, which enters the tricarboxylic acid (TCA) cycle and is converted into citrate. This compound is then transformed into acetyl-CoA by ATP-citrate lyase [41]. The end product of FASN activity is mainly palmitoyl-CoA C16:0, with myristic acid C14:0 present in smaller amounts and stearoyl-CoA C18:0 and lauroyl-CoA C12:0 present in minimal amounts [15].

Research has shown that the expression of FASN is upregulated in glioblastoma tumors compared to normal brain tissue [12,13,42,43]. Moreover, FASN expression is higher in glioblastoma tumors than in low-grade gliomas [12,13,42]. Isocitrate dehydrogenase isoform 1 (*IDH1)* mutation may also result in increased FASN expression [44]. Interestingly, FASN expression varies within glioblastoma tumors depending on the type of cancer cells present. Specifically, higher expression of FASN and increased de novo synthesis of fatty acids have been detected in glioblastoma cancer stem cells [45], which play a crucial role in the stemness and migration of these cells.

Glioblastoma cancer stem cells have been found to secrete extracellular vesicles containing FASN, which are detectable in the blood of glioblastoma patients and, therefore, may serve as a disease marker [43,46]. Due to the significant role of FASN in glioblastoma, FASN inhibitors have demonstrated antitumor properties in vitro and in vivo studies [12,13,14]. Inhibitors of FASN are known to impede glioblastoma cell growth, reduce cell viability, and inhibit the cell cycle, leading to apoptosis and necrosis of glioblastoma cells [12,47]. Additionally, FASN inhibition reduces the growth and stemness of glioblastoma cancer stem cells [13,45]. Studies on animal models have shown that FASN inhibitors can significantly reduce tumor growth, particularly by decreasing angiogenesis through the downregulation of hypoxia-inducible factor-1α (HIF-1α) and VEGF-A [13]. Therefore, FASN inhibitors have prolonged overall survival in mice with intracranial glioblastoma cells. However, according to GEPIA, higher FASN expression does not affect patient prognosis in glioblastoma [48].

Studies have shown that ACC expression is lower in glioblastoma tumors compared to nontumor brain tissue (Table 1) [49]. However, according to the GEPIA database, the expression level of ACC in glioblastoma tumors is not significantly different from healthy brain tissue, and ACC expression does not affect the prognosis for glioblastoma patients [48]. Nonetheless, this does not imply that ACC is not involved in the tumorigenic processes in glioblastoma. The de novo synthesis of fatty acids, particularly through ACC, is crucial for glioblastoma cancer cell proliferation [50]. Inhibition of ACC activity reduces proliferation and induces apoptosis of glioblastoma cancer cells, especially those with epidermal growth factor receptor (EGFR) variant III (EGFRvIII). Additionally, ACC, particularly in the de novo synthesis of fatty acids, is essential for invadopodia formation and, thus, for the migration of glioblastoma cancer cells [51]. Therefore, inhibiting ACC expression and activity can hamper glioblastoma cancer cell migration.

#### 2.2.2. Elongases in Glioblastoma

The expression of Elovl1 in glioblastoma tumors may decrease [52], remain at the same level [49], or, according to the analysis conducted on the GEPIA portal, increase [48] when compared to healthy brain tissue. However, Elovl1 expression in male patients is lower, whereas, in female patients, it is higher in glioblastoma tumors than in healthy tissue [52]. Elovl1 is involved in the elongation of behenoyl-CoA C22:0 to cerotoyl-CoA C26:0 [22]. Notably, a lower level of behenic acid C22:0 has been detected in glioblastoma tumors compared to low-grade gliomas [53]. This could suggest that the expression of Elovl1 may increase in glioblastoma tumors, resulting in a reduction in the substrate for this elongase in the glioblastoma tumor. The expression level of Elovl1 is also related to patient prognosis; according to the GEPIA portal, higher Elovl1 expression in glioblastoma tumors is associated with a poorer prognosis [48].

The expression of Elovl2 in glioblastoma tumors may increase [48,54] or remain at a similar level to that in healthy brain tissue [49,52]. Interestingly, the expression of Elovl2 in glioblastoma tumors may be lower in female patients than in male patients [52]. Elovl2 expression differs depending on the glioblastoma cell type. This protein is expressed in glioblastoma stem-like cells [55], and it is essential for the self-renewal of these cancer stem cells. Moreover, Elovl2 is necessary for EGFR functioning in these cells—it alters the fatty acid composition of lipids in the cell membrane, thus facilitating EGFR activation. In differentiated glioblastoma cells, the expression of Elovl2 is low, and it does not play any significant role in these cells. In vivo studies have shown that decreasing Elovl2 expression in glioblastoma tumors inhibits tumor growth [55]. Studies on glioblastoma patients have also demonstrated the crucial role of Elovl2 in the aggressiveness of this tumor. Patients with higher Elovl2 expression in their tumors exhibited poorer survival than those with lower Elovl2 expression in their tumors compared to healthy tissue [54]. However, it is not conclusive as the analysis on the GEPIA portal did not link the expression level of Elovl2 to patient prognosis [48].

The expression of Elovl3 does not change in glioblastoma tumors compared to the brain tissue without a tumor [48,49,52]. According to the GEPIA portal, higher expression of Elovl3 is associated with a worse prognosis for glioblastoma patients [48], as is the case for Elovl1. Both elongases have been shown to have the same impact on the survival of patients with hepatocellular carcinoma [56]. Elovl3 is significant in the elongation of linoleoyl-CoA C18:2n-6, α-linolenoyl-CoA C18:3n-3, oleoyl-CoA C18:1n-9, and saturated fatty acyl-CoA to behenoyl-CoA C22:0 [22]. Meanwhile, Elovl1 catalyzes the elongation reaction of behenoyl-CoA C22:0 to cerotoyl-CoA C26:0 [22]. This suggests that very long-chain SFAs may be significant in glioblastoma aggressiveness. However, there is a lack of precise studies on this topic.

It has been demonstrated that the expression of Elovl4 remains unchanged in glioblastoma tumors compared to nontumor brain tissue [48,49,52]. However, in males, there is an increase in Elovl4 expression in glioblastoma tumors [52], which does not affect patient prognosis [48].

In contrast, the expression of Elovl5 is elevated in glioblastoma tumors compared to nontumor brain tissue [48,49], although our study did not show this relationship [52]. Additionally, the expression of this elongase is reduced in female glioblastoma tumors [52]. Bioinformatic analysis using the GEPIA portal did not demonstrate a correlation between Elovl5 expression in glioblastoma tumors and patient prognosis [48]. Elovl5 may have both protumor and antitumor properties in glioblastoma. This elongase is involved in the synthesis of ARA 20:4n-6 and EPA C20:5n-3, which are PUFAs that are directly converted to prostanoids by cyclooxygenase-2 (COX-2) [57]. COX-2 expression is elevated in glioblastoma tumors compared to nontumor brain tissue [58]. The most important products of COX-2 activity in cancer processes are prostaglandin E_2_ (PGE_2_) [59] and prostaglandin E_3_ (PGE_3_) [60], which are derived from ARA 20:4n-6 and EPA C20:5n-3, respectively. PGE_2_ is important in the development of glioblastoma tumors [61,62,63] and is associated with radiation resistance [62,64] and TMZ resistance in glioblastoma [65]. On the other hand, PGE_3_ acts as an antitumor agent by reducing the activity of PGE_2_ [60]. Thus, Elovl5, through the production of EPA C20:5n-3 and PGE_3_, partly exhibits antitumor properties. However, high Elovl5 expression tends to result in worse outcomes [48], suggesting the protumor properties of Elovl5, particularly its role in the production of PGE_2_.

Taking into account all patients, the expression of Elovl6 does not change in the glioblastoma tumor compared to the nontumor brain tissue [49,52]. This has also been confirmed by bioinformatic analyses performed on the GEPIA portal [48]. However, the expression of this elongase is decreased in the glioblastoma tumor in women [52]. In murine glioblastoma tumors, Elovl6 expression is increased [66]. Elovl6 is an enzyme that catalyzes the elongation of palmitoyl-CoA to stearoyl-CoA [17]. Therefore, this enzyme is responsible for the synthesis of all SFAs and MUFAs, which are the building blocks of all cells, including glioblastoma cells. Thus, this enzyme is important for the proliferation of glioblastoma cells and, therefore, the growth of these cells. In particular, de novo fatty acid synthesis is increased in glioblastoma cancer stem cells [45], which is important for glioblastoma cancer stem cell stemness. However, there is a lack of studies on the significance of Elovl6 in glioblastoma tumor processes. It seems that this elongase may not be significant in glioblastoma tumor processes. According to GEPIA, high expression of Elovl6 does not affect patient prognosis in glioblastoma [48].

It has been found that the expression of Elovl7 is decreased in the glioblastoma tumor compared to the nontumor brain tissue (Table 2) [52]. These results confirmed the findings of another research team [49] and bioinformatic analyses on the GEPIA portal [48]. However, this effect may depend on sex, as only in women was a decrease in Elovl7 expression in the tumor observed [52]. The expression level of Elovl7 in the glioblastoma tumor, according to the GEPIA portal, does not affect patient prognosis [48]. Elovl7 may be important in the elongation of linoleic acid C18:2n-6 and α-linolenic acid C18:3n-3, where this elongase has the same activity as Elovl3 and Elovl5 [22]. Therefore, Elovl7 participates in the biosynthesis pathway of the n-3 PUFA series. In other stages of fatty acid synthesis, the activity of this elongase is lower than that of other elongases. The 20-carbon n-3 PUFA series has anticancer properties [60,67], which may explain the decrease in Elovl7 expression in the glioblastoma tumor as a result of tumor processes.

#### 2.2.3. Desaturases in Glioblastoma

The next enzymes involved in fatty acid biosynthesis are desaturases, which introduce a double bond into the hydrocarbon chain of acyl-CoA. These enzymes include SCD, SCD5, FADS1, FADS2, and FADS3. SCD expression is reduced in glioblastoma tumors [49,68,69] and is often undetectable. This is due to frequent loss of heterozygosity (LOH), which results in deletions of fragments of chromosome 10 containing the *SCD* and *PTEN* genes [69]. Another cause of reduced SCD expression in glioblastoma is the hypermethylation of DNA fragments responsible for regulating the expression of this enzyme [69]. However, it appears that reduced SCD expression in glioblastoma tumors does not affect the composition of fatty acids in the tumor. In glioblastoma tumors, there is no change in the levels of palmitic acid C16:0, stearic acid C18:0, oleic acid C18:1n-9, or palmitoleic acid C16:1n-7 compared to low-grade glioma [53] or brain tissue without tumors [70].

SCD expression may occur in glioblastoma cancer cells, as shown by analyses of various cell lines [69]. SCD expression in glioblastoma tumors may be locally increased, as it is increased by activation of EGFR [71] and platelet-derived growth factor receptor (PDGFR) [72], which activate sterol regulatory element-binding protein 1 (SREBP1) through these receptors. SCD expression in glioblastoma cancer cells may be increased by endoplasmic reticulum (ER) stress [73]. Hypoxia has been shown to increase SCD expression through SREBP-1 [74], and *IDH1* mutation increases SCD expression through the oncometabolite D-2-hydroxyglutarate, mainly in low-grade glioma and secondary glioblastoma where these mutations are common [75,76]. Isolated cell lines from glioblastoma tumors have diverse SCD expression [69], some lines showing no detectable SCD expression, and others with high expression. Increased or high expression of SCD is associated with the growth and proliferation of glioblastoma cells [69].

SCD is also important for the functioning of glioblastoma cancer stem cells, particularly for their stemness [73], and is responsible for TMZ resistance [77]. This is related to the activation of the Akt/protein kinase B (PKB) → glycogen synthase kinase 3β (GSK3β) → β-catenin pathway. Inhibition of SCD also results in a decrease in MUFA and an accumulation of SFA in glioblastoma cells [73], leading to ER stress and apoptosis of glioblastoma cancer cells through activation of inositol-requiring enzyme 1 (IRE1) and c-Jun N-terminal kinase (JNK) mitogen-activated protein kinase (MAPK). SFA accumulation also inhibits RAD51-dependent DNA repair [73]. Therefore, SCD inhibitors increase the susceptibility of glioblastoma cells to TMZ. Consequently, SCD inhibitors may be effective in treating glioblastoma if there is a high expression of this enzyme in the glioblastoma tumor, as shown by experiments in mice [69,73]. According to the GEPIA database, the level of SCD expression in glioblastoma tumors does not affect patient prognosis [48], which may indicate a limited role of this enzyme in the tumorigenesis of glioblastoma.

The expression of SCD5 in glioblastoma tumors does not differ from that of nontumor brain tissue [49,68]. However, according to data from the GEPIA portal, the expression of SCD5 is elevated in glioblastoma tumors [48]. In vitro studies have not shown the significant importance of this enzyme in the viability of various glioblastoma cell lines [69]. Additionally, the level of SCD5 expression in glioblastoma tumors is not associated with patient prognosis, as reported by the GEPIA portal [48].

The expression level of FADS1 in glioblastoma tumors does not differ from that of nontumor brain tissue [48,49,68]. However, FADS1 expression may depend on the type of glioblastoma cell. In glioblastoma cancer stem cells, FADS1 expression is higher than in nonstem cancer cells, and its activity is critical for cell viability and self-renewal [78]. Nevertheless, the level of FADS1 expression in glioblastoma tumors is not associated with patient prognosis, according to the GEPIA portal [48].

The expression of FADS2 is elevated in glioblastoma tumors compared to nontumor brain tissue [48,49]. However, our results showed the opposite relationship [68]. FADS2 expression in glioblastoma cancer cells is upregulated following PDGFR activation [72]. Similar to FADS1, FADS2 expression may depend on the type of glioblastoma cell. In glioblastoma cancer stem cells, FADS2 expression is higher than in nonstem cancer cells, and its activity is critical for cell viability and self-renewal [78]. Furthermore, FADS2 activity leads to greater glioblastoma tumor growth and radioresistance [79], which is associated with the production of ARA C20:4n-6. This PUFA is converted to PGE_2_ by COX-2, and this lipid mediator is responsible for radioresistance. Therefore, FADS2/D6D inhibitors such as SC-26196 exhibit in vivo antitumor activity against glioblastoma [55,79]. However, increased FADS2 expression is not associated with patient prognosis, according to the GEPIA portal [48].

The expression of FADS3 in glioblastoma tumors does not differ from that of nontumor brain tissue (Table 3) [48,49,68]. However, FADS3 is one of the few fatty acid synthesis genes whose increased expression in glioblastoma tumors is associated with worse patient prognosis, according to the GEPIA portal [48].

#### 2.2.4. DHA, EGFR, and Glioblastoma

DHA C22:6n-3 in glioblastoma tumors is significant in the process of tumorigenesis. The production of this PUFA mainly occurs in glioblastoma cancer stem cells, which possess higher expression of Elovl2 than glioblastoma cancer nonstem cells [55]. Elovl2 is responsible for the elongation of EPA C20:5n-3 in the PUFA biosynthesis pathway and the production of DHA C22:6n-3 from this PUFA.

DHA C22:6n-3 is essential for the proliferation and self-renewal of glioblastoma cancer stem cells, which is related to the functioning of lipid rafts and EGFR. DHA C22:6n-3 is significant in the formation of lipid rafts, which are microdomains enriched in cholesterol and sphingolipids and are located in the fluid cell membrane.

EGFR is an example of a membrane protein found in lipid rafts, and upon its activation, changes occur in the cell membrane structure via phospholipase D_2_ (PLD_2_), leading to the formation of nanoclusters in the cell membrane in close proximity to the receptor, which is necessary for signal transduction [80]. DHA C22:6n-3 increases the amount of EGFR in lipid rafts and thus facilitates signal transduction from this receptor [81]. This function of DHA C22:6n-3 is observed at concentrations of a few micromoles [81], which is the concentration at which this PUFA occurs in the brain in the unesterified form [82].

At higher concentrations of several tens of micromoles, DHA C22:6n-3 is incorporated and builds glycerophospholipids, which are then part of the lipid rafts [83]. At high concentrations, DHA C22:6n-3 reduces membrane rigidity [84] and disturbs the structure of lipid rafts, causing the internalization of EGFR along with lipid rafts and their degradation in lysosomal pathways [85,86]. This results in a decrease in the amount of EGFR in the cell membrane [87]. Moreover, DHA C22:6n-3 at high concentrations decreases the amounts of sphingomyelin and cholesterol in lipid rafts [83] and hinders EGFR clustering by disrupting cholesterol-EGFR interactions [84]. Since lipid rafts and the interaction of lipids with EGFR are necessary for signal transduction from this receptor, the activity of EGFR is reduced [84,87]. As a result of DHA C22:6n-3’s actions at high concentrations, processes dependent on this receptor are blocked, resulting in the inhibition of wound healing [88]. Additionally, EGFR is significant in the functioning of certain cancers. Therefore, DHA C22:6n-3 at high concentrations, by inhibiting EGFR activity, causes apoptosis of cancer cells, as demonstrated in breast cancer cells [87] and pancreatic cancer cells [89].

Based on the information presented in this section, it can be inferred that exogenous supplementation of DHA C22:6n-3 may have anticancer properties against glioblastoma. However, endogenous production of this PUFA has procancer properties. Therefore, blocking the biosynthesis of DHA C22:6n-3, for example, by inhibiting Elovl2, will also have anticancer properties related to the disruption of EGFR function [55].

#### 2.2.5. The Effect of Hypoxia on Fatty Acid Synthesis in Glioblastoma

Studies on U87 MG cells have shown that hypoxia increases the expression of SCD [74], as well as the expression of elongases Elovl5 and Elovl6 [52]. However, hypoxia decreases the expression of FASN and ACC [74], as well as the expression of elongases Elovl1, Elovl3, Elovl4, and Elovl7 [52]. FASN is responsible for the de novo production of SFA and MUFA [23]. SCD is responsible for the production of MUFA, while Elovl1, Elovl3, and Elovl7 are responsible for the elongation of saturated fatty acyl-CoA, i.e., for the production of SFA. Meanwhile, the activity of Elovl5 is important in the biosynthesis pathway of PUFA. Therefore, hypoxia reduces the overall production of de novo fatty acids, particularly SFA, but increases the production of MUFA and PUFA. This leads to an increase in the fluidity of the cell membrane, which may have significant implications for the transduction of signals from membrane receptors, such as EGFR and PDGFR [55].

#### 2.2.6. The Impact of IDH1 Mutation on Fatty Acid Synthesis

*IDH1* mutations are characteristic of low-grade glioma and secondary glioblastoma, where depending on the type of tumor, 30% to 85% of tumors have this mutation [76]. In primary glioblastoma, less than 10% of tumors have a mutation in *IDH1* [75,76]. *IDH1* mutations are also found in leukemias, such as acute myeloid leukemia (AML) [90,91] and acute lymphoblastic leukemia (ALL) [92]. Mutations in the *IDH1* gene lead to changes in the catalytic activity of the *IDH1* enzyme. This enzyme begins to produce the oncometabolite D-2-hydroxyglutarate from α-ketoglutarate [93]. This compound is an inhibitor of 2-oxoglutarate-dependent dioxygenases, which participate in histone and DNA demethylation. Therefore, *IDH1* mutations lead to hypermethylation of DNA and histones, resulting in epigenetic changes in DNA that alter the expression of many genes. Specifically, there is an increase in the expression of FASN and SCD [44]. Higher expression of FASN and SCD in cells with *IDH1* mutations leads to increased production and accumulation of MUFA in intracellular membranes [44]. This leads to changes in the morphology of the endoplasmic reticulum and Golgi apparatus, particularly the dilation of these organelles.

## 3. Synthesis of Glycerophospholipids and Glioblastoma Tumorigenesis

### 3.1. Glycerol-3-Phosphate Acyltransferases and Glioblastoma Tumorigenesis

#### 3.1.1. Glycerol-3-Phosphate Acyltransferases

Fatty acyl-CoAs are utilized in the synthesis of glycerophospholipids and triacylglycerol (TAG). In the first step of this pathway, the acyl group is transferred from acyl-CoA to the *sn*-1 position of glycerol-3-phosphate, producing 1-acylglycerol-3-phosphate (also known as lysophosphatidic acid). The enzymes responsible for this reaction are glycerol-3-phosphate acyltransferases (GPATs) (Figure 2) [94]. Humans have four GPAT isoforms: GPAT1/*GPAM*, GPAT2, 1-acylglycerol-3-phosphate O-acyltransferase (AGPAT)10/AGPAT9/GPAT3, and GPAT4. However, the activity of GPAT3 is disputed [95], as it also has AGPAT activity and is therefore referred to as AGPAT10 [95]. Due to inconsistencies in the nomenclature of AGPAT9 and AGPAT10, this enzyme is referred to as AGPAT10/AGPAT9/GPAT3 in this work. AGPAT6/GPAT4 also exhibits AGPAT activity and is also known as AGPAT6 [96,97].

#### 3.1.2. Glycerol-3-Phosphate Acyltransferases in Glioblastoma

Currently, according to the PubMed browser (https://pubmed.ncbi.nlm.nih.gov, accessed on 5 November 2022), there are no articles available investigating GPAT in glioma or glioblastoma. According to the GEPIA portal, there are no changes in the expression of GPAT1, GPAT2, and GPAT3/AGPAT10 in glioblastoma tumor tissue compared to healthy brain tissue (Table 4) [48]. Additionally, this same portal does not associate patient prognosis in glioblastoma with the expression of GPAT1 and GPAT2. However, according to GEPIA, higher expression of AGPAT10/AGPAT9/GPAT3 in glioblastoma tumor tissue is associated with a worse prognosis [48]. Furthermore, there is higher expression of AGPAT6/GPAT4 in glioblastoma tumor tissue than in healthy brain tissue [48]. However, transcriptomic analysis performed by Seifert et al. showed that there is no higher expression of GPAT1 and AGPAT6/GPAT4 in glioblastoma tumor tissue compared to healthy brain tissue [49].

### 3.2. Dihydroxyacetone Phosphate Pathway and Glioblastoma Tumorigenesis

#### 3.2.1. Dihydroxyacetone Phosphate Pathway

An alternative pathway in the de novo synthesis of glycerophospholipids and TAG involves the use of dihydroxyacetone phosphate instead of glycerol-3-phosphate [98,99,100]. This pathway is mainly responsible for ether lipid biosynthesis [101]. Dihydroxyacetone phosphate is converted to 1-acyl-dihydroxyacetone 3-phosphate and acyl-CoA by dihydroxyacetone phosphate acyltransferase (DHAPAT)/GNPAT [102]. Subsequently, 1-acyl-dihydroxyacetone-3-phosphate is reduced by the enzyme with 1-acyl-dihydroxyacetone-3-phosphate reductase activity to produce 1-acylglycerol-3-phosphate [103].

#### 3.2.2. Dihydroxyacetone Phosphate Pathway in Glioblastoma

According to the GEPIA database, the expression level of DHAPAT/GNPAT in glioblastoma tumors does not differ from that of healthy brain tissue, and the expression of this gene in glioblastoma does not affect patient prognosis [48]. However, a transcriptomics analysis by Seifert et al. showed higher DHAPAT/GNPAT expression in glioblastoma compared to healthy brain tissue [49]. The significance of this enzyme in glioblastoma tumorigenesis requires further investigation.

### 3.3. 1-Acylglycerol-3-phosphate O-acyltransferases and Glioblastoma Tumorigenesis

#### 3.3.1. 1-Acylglycerol-3-phosphate O-acyltransferases

In the next step, phosphatidic acid (1,2-diacylglycerol-3-phosphate) is generated from lysophosphatidic acid (also known as 1-acylglycerol-3-phosphate) and one acyl-CoA by AGPAT (Figure 3) [97]. There are 11 isoforms of AGPAT: AGPAT1-11. These enzymes introduce the fatty acid residue at the *sn*-2 position in 1-acylglycerol-3-phosphate. AGPATs show the highest substrate specificity for oleoyl-CoA C18:1n-9 [97,104,105,106]. However, AGPAT8/acyl-CoA:lysocardiolipin acyltransferase (ALCAT)1/lysocardiolipin acyltransferase (LCLAT)1 shows the highest specificity for palmitoyl-CoA C16:0 and oleoyl-CoA C18:1n-9 at a similar level [107]. Some of the enzymes in this group also exhibit lysophospholipids acyltransferase (LPLAT) activity. In particular, this activity, important for remodeling the composition of cell membranes and intracellular membranes, is exhibited by AGPAT3 [106], AGPAT5 [106], AGPAT9/lysophosphatidylcholine acyltransferase (LPCAT)1 [108], and AGPAT11/LPCAT2 [105]. These enzymes are located in the endoplasmic reticulum [97,106].

#### 3.3.2. 1-Acylglycerol-3-phosphate O-acyltransferases in Glioblastoma

According to GEPIA, the expression of certain AGPATs is increased in glioblastoma tumors compared to healthy brain tissue. In particular, AGPAT5, AGPAT6/GPAT4, AGPAT8/ALCAT1/LCLAT1, AGPAT9/LPCAT1, and AGPAT11/LPCAT2 show higher expression in glioblastoma tumors than in healthy tissue [48]. However, the expression of AGPAT7/LPEAT2/LPCAT4 is lower in glioblastoma tumors than in healthy brain tissue [48]. The expression levels of AGPAT1-4 and AGPAT10/AGPAT9/GPAT3 do not differ between glioblastoma tumors and healthy brain tissue. A transcriptomic analysis by Seifert et al. showed that the expression of AGPAT5 and AGPAT9/LPCAT1 is higher in glioblastoma tumors (as in GEPIA), but the expression of AGPAT3, AGPAT4, and AGPAT7/LPEAT2/LPCAT4 (as in GEPIA) is lower [49]. The expression of AGPAT1, AGPAT2, AGPAT6/GPAT4, AGPAT8/ALCAT1/LCLAT1, and AGPAT11/LPCAT2 remains unchanged. According to the GEPIA portal, higher expression of AGPAT10/AGPAT9/GPAT3 in glioblastoma tumors is associated with a worse prognosis for the patient [48]. The expression levels of other AGPATs are not associated with patient prognosis in glioblastoma.

AGPAT7/LPEAT2 is important in remodeling glycerophospholipids in intracellular membranes by introducing DHA C22:6n-3 into lysophospholipids [109]. The lower expression in glioblastoma tumors indicates that this PUFA is less intensively incorporated into glycerophospholipids and TAGs than in healthy brain tissue.

In glioblastoma, there is increased expression of AGPAT5, AGPAT9/LPCAT1, and AGPAT11/LPCAT2 compared to healthy brain tissue (Table 5) [48]. These enzymes are also lysophospholipid acyltransferases and participate in the Lands cycle, a process involving the re-esterification of phospholipids [105,106,108]. In addition, higher expression of phospholipase A_2_ (PLA_2_), which removes fatty acids from the *sn*-2 position of phospholipids, has been observed in glioblastoma [48,49,110]. Specifically, cytosolic phospholipase A_2_ (cPLA_2_)α/PLA2G4A and various sPLA_2_, including PLA2G5, show increased expression in glioblastoma [48,49,111]. Moreover, higher expression of calcium-independent phospholipase A_2_ (iPLA_2_)η/PNPLA4 in glioblastoma is associated with a worse prognosis for the patient [48]. The higher expression of enzymes that remove and introduce fatty acids at the *sn*-2 position of phospholipids suggests that intensive re-esterification of phospholipids is taking place in glioblastoma, i.e., the Lands cycle.

### 3.4. Synthesis of Glycerophospholipids and Triacylglycerol from Phosphatidic Acid

Phosphatidic acid (1,2-diacylglycerol-3-phosphate) serves as a precursor for the synthesis of glycerophospholipids and TAGs (Figure 4). This pathway bifurcates phosphatidic acid into two distinct pathways [100]. Phosphatidic acid is converted to diacylglycerol (DAG) by enzymes with phosphatidate phosphatase activity, or it is transformed into cytidine diphosphate-diacylglycerol (CDP-DAG) by enzymes with CDP-DAG synthase activity [112]. DAG gives rise to TAG, phosphatidylcholine (PC), phosphatidylethanolamine (PE), and phosphatidylserine (PS), while CDP-DAG gives rise to phosphatidylinositol (PI), phosphatidylglycerol (PG), and cardiolipin (CL).

### 3.5. Lipins and Glioblastoma Tumorigenesis

#### 3.5.1. Lipins

In the next stage of glycerophospholipid and TAG biosynthesis, the phosphate group is removed from phosphatidic acid (1,2-diacylglycerol-3-phosphate) to produce DAG (Figure 5). This reaction is catalyzed by enzymes with phosphatidate phosphatase activity. The enzymes responsible for this reaction in lipid synthesis are lipins. There are three isoforms of lipin: lipin 1, lipin 2, and lipin 3 [113], which catalyze the same reaction but differ in terms of their expression locations [114]. In humans, lipin 1 and lipin 2 are expressed in the brain, lipin 2 in the liver, lipin 1 in muscles, lipin 1 and lipin 2 in the adipose tissue, and lipin 3 in the gastrointestinal tract [114].

#### 3.5.2. Lipins in Glioblastoma

According to the GEPIA portal, the expression of lipins does not differ between glioblastoma tumors and healthy brain tissue (Table 6) [48]. However, the analysis of transcriptomics by Seifert et al. showed that the expression level of lipin 1 is lower in the tumor, while lipin 2 and lipin 3 do not differ from healthy brain tissue [49]. Moreover, according to GEPIA, the expression level of these enzymes does not significantly affect the prognosis for patients with glioblastoma [48]. This suggests that these proteins may not play a significant role in the tumor processes in glioblastoma. However, given the function of these enzymes in other models, lipins may be essential in increasing TAG production and the formation of lipid droplets in glioblastoma cancer cells under the influence of hypoxia. Lipins may also affect tumor processes in glioblastoma by influencing peroxisome proliferator-activated receptors (PPARs) and, in the case of lipin 2, P2X7.

### 3.6. Biosynthesis of Phosphatidylethanolamine and Glioblastoma Tumorigenesis

#### 3.6.1. Biosynthesis of Phosphatidylethanolamine

DAG is converted to PE via the Kennedy pathway [118,119]. In the first reaction, ethanolamine is phosphorylated to phosphoethanolamine by ethanolamine kinase. There are two enzymes with this activity: ethanolamine kinase (ETNK)1 [120] and ETNK2 [121]. In the next reaction of the Kennedy pathway, phosphoethanolamine and cytidine triphosphate (CTP) are converted to CDP-ethanolamine by CTP:phosphoethanolamine cytidylyltransferase (ECT)/phosphate cytidylyltransferase (PCYT)2 [122,123].

In the final reaction, the enzyme CDP-ethanolamine:1,2-diacylglycerol ethanolaminephosphotransferase catalyzes the formation of PE (Figure 6) [119]. The enzyme responsible for this reaction is choline/ethanolaminephosphotransferase (CEPT1) [124,125] and selenoprotein I (SELENOI) (formerly known as ethanolaminephosphotransferase 1 (EPT1)) [126]. CEPT1 is an enzyme that exhibits both ethanolaminephosphotransferase and cholinephosphotransferase activity [124]. However, this enzyme shows greater cholinephosphotransferase activity than ethanolaminephosphotransferase activity [124].

In addition to the Kennedy pathway, PE can be synthesized from PS with the involvement of phosphatidylserine decarboxylase (PISD)/PSD [127]. This is a mitochondrial enzyme located in the inner mitochondrial membrane [128]. The synthesis of PE in mitochondria via this pathway is essential for the proper functioning of these organelles [129,130].

#### 3.6.2. Biosynthesis of Phosphatidylethanolamine in Glioblastoma

In glioma, a mutation in the *IDH1* gene affects the synthesis of PE. *IDH1* mutation results in the production of 2-hydroxyglutarate, which increases the activation of HIF-1 [131]. This transcription factor reduces the expression of ETNK2 and thereby reduces the production of PE. However, *IDH1* mutation is common in low-grade glioma [76]. In contrast, in primary glioblastoma, less than 10% of tumors have *IDH1* mutation [75,76].

According to GEPIA, the expression of enzymes involved in PE biosynthesis affects prognosis. Higher expression of ETNK1 is associated with a better prognosis [48]. Conversely, higher expression of ETNK2 in glioblastoma is associated with a worse prognosis for patients with this type of cancer. However, according to GEPIA and Seifert et al., the expression of both enzymes in glioblastoma does not differ from healthy brain tissue [48,49]. Both enzymes catalyze the same biochemical reaction, but their expression has the opposite effect on survival. These enzymes differ in that ETNK2 also has slight choline kinase activity [121]. However, according to the GEPIA database, the enzymes with choline kinase activity, choline kinase (CHK)α and CHKβ, do not affect the prognosis for glioblastoma patients [48]. Therefore, this difference cannot explain the observed differences in prognosis.

A similar relationship has been demonstrated in gastric cancer. It is suggested that miR-199a-3p, by lowering ETNK1 expression, has a protumor effect in gastric cancer [132]. On the other hand, ETNK2 in gastric cancer has an antiapoptotic effect, increases proliferation and migration, and causes metastasis of gastric cancer to the liver [133]. Higher expression of ETNK2 in gastric cancer is associated with a worse prognosis. Therefore, ETNK2 in gastric cancer is an oncogene, as it is in glioblastoma. Analysis of the GEPIA platform of other types of tumors showed that higher expression of ETNK2 is associated with a worse prognosis for patients with acute myeloid leukemia. However, in low-grade glioma, higher expression of ETNK1, like ETNK2, is associated with a worse prognosis. In kidney renal clear cell carcinoma, higher expression of ETNK1, like ETNK2, is associated with a better prognosis [48]. Therefore, the significance of both enzymes in cancer processes depends on the type of tumor and requires further investigation, especially in the mechanisms of glioblastoma.

According to the GEPIA platform, the expression of PISD, ECT/PCYT2, and SELENOI does not differ between glioblastoma tumor tissue and healthy brain tissue (Table 7) [48]. However, CEPT1 is expressed at a higher level in glioblastoma tumor tissue compared to healthy brain tissue, and the expression of these enzymes is not associated with patient prognosis. A trend towards better prognosis (*p* = 0.062) was observed with higher CEPT1 expression in glioblastoma tumors [48]. In contrast, Seifert et al. (2015) found that the expression of ECT/PCYT2 and SELENOI is decreased, while the expression of CEPT1 and PISD is unchanged in glioblastoma tumors compared to healthy brain tissue [49].

The higher expression of CEPT1 in glioblastoma tumors indicates greater biosynthesis of PE and PC than in healthy brain tissue, which explains the higher levels of these glycerophospholipids in glioblastoma tumors compared to healthy brain tissue. Additionally, nonstem cells in glioblastoma tumors contain more PE than cancer stem cells, indicating higher biosynthesis of this glycerophospholipid in glioblastoma tumors, particularly in nonstem cells [78]. The association between higher CEPT1 expression and better patient outcomes in glioblastoma can be explained by lipid metabolism. DAG is used to produce PE, PC, and TAG. Increased CEPT1 expression leads to increased production of PE and PC from DAG and decreased production of TAG. Greater production of TAG is associated with poorer outcomes for glioblastoma patients [134,135]. Higher expression of DGAT1, the enzyme directly involved in TAG biosynthesis [135], as well as lipid droplets, which store TAG [134], is associated with worse outcomes.

### 3.7. Biosynthesis of Phosphatidylcholine and Glioblastoma Tumorigenesis

#### 3.7.1. Biosynthesis of Phosphatidylcholine

DAG is transformed into PC in the Kennedy pathway, similar to the biosynthesis of PE discussed previously [118,119]. In the first reaction, choline is phosphorylated by choline kinase, resulting in the formation of phosphocholine. Two choline kinases have been identified: CHKα/*CHKA* [136,137,138] and CHKβ/*CHKB* [138]. In the next reaction, CDP-choline is formed from phosphocholine and CTP by CTP:phosphocholine cytidylyltransferase. Two enzymes with this activity have been identified: CTP:phosphocholine cytidylyltransferase α (CCTα)/*PCYT1A* [139] and CTP:phosphocholine cytidylyltransferase β (CCTβ)/*PCYT1B* [140]. In the final reaction of the Kennedy pathway, PC is formed from CDP-choline and DAG with the participation of an enzyme with cholinephosphotransferase activity [119]. In humans, two enzymes participate in this stage of PC biosynthesis: CEPT1 [124] and cholinephosphotransferase 1 (CHPT1) [141]. CEPT1 exhibits both cholinephosphotransferase and ethanolaminephosphotransferase activity [124,125]. On the other hand, CHPT1 exhibits only cholinephosphotransferase activity and not ethanolaminephosphotransferase activity [141,142].

The Kennedy pathway is not the only route for PC biosynthesis. This glycerophospholipid can be formed from PE by phosphatidylethanolamine N-methyltransferase (PEMT) (Figure 7) [143,144,145]. However, these two pathways synthesize PC that differ from each other. The Kennedy pathway mainly produces PC composed of palmitic acid C16:0 and oleic acid C18:1n-9 [146]. On the other hand, PC produced by PEMT have significantly more PUFAs, especially ARA C20:4n-6 and DHA C22:6n-3 [146,147]. This could be significant in the functioning of cell membranes and intracellular membranes as well as in the functions performed by these PUFAs.

#### 3.7.2. Biosynthesis of Phosphatidylcholine in Glioblastoma

The expression of CHKα is lower in glioblastoma tumors compared to healthy brain tissue [48,148]. However, Seifert et al. (2015) do not confirm this finding [49]. On the other hand, the expression of CHKβ in glioblastoma tumors does not differ from that in healthy brain tissue [48,148].

CCTβ/PCYT1B expression has been reported to be lower in glioblastoma and anaplastic astrocytomas than in healthy brain tissue [148]. However, Seifert et al. and GEPIA do not confirm this in glioblastoma tumors [48,49]. Meanwhile, the expression of CCTα/PCYT1A and CHPT1 does not differ from that in healthy brain tissue [48,148]. However, Seifert et al. (2015) found that CHPT1 expression is elevated in glioblastoma tumors [49].

CEPT1 expression is reported to be higher in glioblastoma tumors than in healthy brain tissue by GEPIA [48], but Seifert et al. (2015) do not confirm this finding [49]. PEMT expression, on the other hand, is consistently reported to be elevated in glioblastoma tumors compared to healthy brain tissue [48,49].

The expression levels of these enzymes in glioblastoma tumors do not significantly affect prognosis. However, GEPIA found a trend of better prognosis (*p* = 0.062) with higher CEPT1 expression in glioblastoma tumors (Table 8) [48].

The elevated expression of PEMT suggests that there is an intensive conversion of PE to PC in glioblastoma tumors. Furthermore, some sources indicate that the expression of certain enzymes involved in the Kennedy pathway of PC synthesis, including CHPT1 and CEPT1, is elevated in glioblastoma tumors. This may explain why there is a higher level of PC in glioblastoma tumors than in healthy brain tissue [150]. However, some sources also suggest that CHKα [48,148] and CCTβ/PCYT1B [148] expression is lower in glioblastoma tumors, which requires further investigation. PEMT is also likely the primary source of PC in these tumors through the conversion of PE, and hence its role in the processes of glioblastoma tumorigenesis needs to be explored further.

CHKα plays a significant role in the tumorigenesis of glioblastoma. It promotes the proliferation and migration of glioblastoma cells by activating Akt/PKB [151,152,153]. CHKα is also important in the migration of glioblastoma cancer cells, as its expression is upregulated during epithelial-mesenchymal transition (EMT) depending on zinc finger E-box binding homeobox 1 (ZEB1) [151]. Additionally, CHKα is essential in the stemness of glioblastoma cancer cells [151]. CHKα2, besides its role in phosphocholine biosynthesis, can also function as a protein kinase. This enzyme can phosphorylate perilipin 2 (PLIN2) at Tyr^232^ and PLIN3 at Tyr^251^ [154]. As a result, PLIN2 and PLIN3 are proteolytically degraded, leading to lipid droplet lipolysis and the utilization of lipids from lipid droplets as a source of energy during nutrient deficiency. This mechanism has been observed in glioblastoma [154]. In summary, CHKα plays a critical role in glioblastoma tumorigenesis, including proliferation, migration, EMT, stemness, and lipid metabolism.

CHKα is also significant in EGFR-induced proliferation, as demonstrated by experiments on a breast cancer model [155]. Upon activation of EGFR, c-Src phosphorylates CHKα at Tyr^197^ and Tyr^333^, which increases CHKα activity and, consequently, the proliferation of cancer cells. The mechanism of EGFR action on CHKα has been observed in breast cancer models, but EGFR also plays an important role in glioblastoma cancer [156]. However, this described mechanism needs to be investigated in glioblastoma.

In glioma, mutations in the *IDH1* gene impact the synthesis of PC. The *IDH1* mutation leads to the production of 2-hydroxyglutarate, which increases the activation of HIF-1 [131]. This transcription factor reduces the expression of CHKα and, thus, PC production. Although the *IDH1* mutation is common in low-grade glioma [76], in primary glioblastoma, less than 10% of tumors have the *IDH1* mutation [75,76].

The significance of other PC synthesis enzymes in glioblastoma cancer has not been studied. Therefore, to better understand glioblastoma, it is necessary to investigate these enzymes, especially PEMT, whose expression is increased in the glioblastoma tumor, and CEPT1, an enzyme that may impact the prognosis for glioblastoma patients.

### 3.8. Biosynthesis of Phosphatidylserine and Glioblastoma Tumorigenesis

#### 3.8.1. Biosynthesis of Phosphatidylserine

PS is synthesized from PC and PE by phosphatidylserine synthases. Two enzymes with this activity have been identified: phosphatidylserine synthase 1 (PTDSS1), which in humans produces PS from both PE and PC [157], and phosphatidylserine synthase 2 (PTDSS2), which produces PS from PE but not from PC [157]. However, these enzymes can also catalyze the reverse reaction (Figure 8) [157,158].

#### 3.8.2. Biosynthesis of Phosphatidylserine in Glioblastoma

In glioblastoma tumors, the level of PS is similar to that in healthy brain tissue (Table 9) [159]. However, there is more PS in glioblastoma cancer nonstem cells than in glioblastoma cancer stem cells [78]. According to GEPIA, there is higher expression of PTDSS1 in glioblastoma tumors than in healthy brain tissue. The expression of PTDSS2 in glioblastoma tumors does not differ from that in healthy brain tissue [48]. However, Seifert et al. reported lower expression of PTDSS1 and higher expression of PTDSS2 in glioblastoma tumors than in healthy brain tissue [49]. Additionally, higher expression of PTDSS2 is associated with worse patient prognosis, according to GEPIA [48].

The significance of PTDSS2 in glioblastoma tumorigenesis may be explained by the role of PS, a phospholipid located in the inner bilayer of the cell membrane. During apoptosis, PS is translocated to the outer layer of the cell membrane and serves as an “eat me” signal for macrophages [160]. It should be noted that cancer cell apoptosis is a part of tumor function [161,162]. Higher expression of phosphatidylserine synthases results in a greater amount of PS in cancer cells, which exposes macrophages to a higher amount of PS during apoptosis [163]. This leads to the polarization of macrophages into M2 macrophages [164], which participate in tumorigenesis. Thus, higher expression of phosphatidylserine synthases increases the amount of PS in cancer cells, which upon apoptosis, more strongly polarizes macrophages into M2 macrophages, leading to intensified tumorigenesis and a worse prognosis for cancer patients.

### 3.9. CDP-DAG Synthases and Glioblastoma Tumorigenesis

#### 3.9.1. CDP-DAG Synthases

CDP-DAG synthases, including CDP-DAG synthase 1 (CDS1) [165], CDP-DAG synthase 2 (CDS2) [166], and TAMM41 [167,168] are enzymes that catalyze the formation of CDP-DAG from phosphatidic acid (also known as 1,2-diacylglycerol-3-phosphate) and CTP. These enzymes differ in their subcellular localization and the specific phospholipids synthesized by the biochemical pathway in which they are involved. CDS1 and CDS2 are located in the endoplasmic reticulum [167,169], where they form a complex with AGPAT2 [170]. This allows for the immediate conversion of 1-acylglycerol-3-phosphate to CDP-DAG, which is then further converted to PI. TAMM41, on the other hand, is found in mitochondria and participates in the synthesis of PG and CL [167,168].

#### 3.9.2. CDP-DAG Synthases in Glioblastoma

Currently, according to the PubMed database (https://pubmed.ncbi.nlm.nih.gov, accessed on 5 November 2022), there are no available articles investigating CDS1, CDS2, or TAMM41 in glioma or glioblastoma. Analyses using the GEPIA portal showed that CDS1 expression is decreased in glioblastoma compared to healthy brain tissue. However, CDS2 and TAMM41 expression does not change (Table 10) [48]. Seifert et al. obtained similar results [49]. Additionally, higher CDS2 expression is associated with a better prognosis for glioblastoma patients, while CDS1 and TAMM41 expression are not correlated with prognosis. However, there are very few studies investigating the differences between these two CDP-DAG synthases. Therefore, the significance of CDS2 as an antioncogene in glioblastoma requires further investigation.

### 3.10. Biosynthesis of Phosphatidylinositol and Phosphatidylinositol Phosphate and Glioblastoma Tumorigenesis

#### 3.10.1. Biosynthesis of Phosphatidylinositol and Phosphatidylinositol Phosphate

CDP-DAG and *myo*-inositol undergo transformation to PI and cytidine monophosphate (CMP) by CDP-diacylglycerol-inositol 3-phosphatidyltransferase (CDIPT), also known as phosphatidylinositol synthase (PIS) [171,172,173]. CDIPT is located in the endoplasmic reticulum [171]. This enzyme has no specificity towards any particular CDP-DAG [174]. After the synthesis of PI, it is phosphorylated by phosphatidylinositol 4-kinases (PI4K), phosphatidylinositol-4-phosphate 5-kinases (PIP5K), and phosphoinositide kinase, FYVE-type zinc finger containing (PIKFYVE) (formerly known as phosphatidylinositol-3-phosphate/phosphatidylinositol 5-kinase, type III) to produce various phosphatidylinositol phosphates [175,176,177]. The most important of the phosphatidylinositol phosphates is phosphatidylinositol-4,5-bis-phosphate (PIP_2_) due to the significant role it plays in cellular physiology (Figure 9).

#### 3.10.2. Biosynthesis of Phosphatidylinositol and Phosphatidylinositol Phosphate in Glioblastoma

According to analyses conducted using the GEPIA portal, the expression of CDIPT does not differ between glioblastoma tumors and healthy brain tissue [48]. However, Seifert et al. demonstrated that CDIPT expression is lower in glioblastoma tumors compared to healthy brain tissue [49]. Additionally, there is a trend (*p* = 0.07) towards a worse prognosis for glioblastoma patients with higher expression of this enzyme, according to GEPIA [48]. Furthermore, differences in the expression of PI4K and PIP5K have been observed between glioblastoma tumors and healthy brain tissue. Specifically, according to GEPIA, glioblastoma tumors show lower expression of the following:Phosphatidylinositol 4-kinase type III, α (PI4KIIIα)/PI4KA;Phosphatidylinositol-4-phosphate 5-kinase, type I, β (PIP5KIβ)/PIP5K1B;PIP5KIγ/PIP5K1C.

Moreover, glioblastoma tumors show elevated expression of PIP5K1A compared to healthy brain tissue. The expression of remaining PI4K and PIP5K in glioblastoma tumors does not differ from healthy brain tissue [48]. However, according to Seifert et al., the expression of PIP4K2A, PIP4K2B, PIP4K2C, PIP5KIβ/PIP5K1B (as well as GEPIA), and PIP5KIγ/PIP5K1C (as well as GEPIA) is reduced in glioblastoma tumors [49]. In contrast, the expression of PI4K2B is increased. Furthermore, according to Seifert et al., the expression of PI4K2A, PI4KB, PIP5K1A, and PIKFYVE remains unchanged [49]. The GEPIA database shows that the expression of PI4K, PIP5K, and PIKFYVE is not significantly associated with patient prognosis in glioblastoma [48]. The level of PI in glioblastoma tumors is higher than in healthy brain tissue [159]. Glioblastoma cancer nonstem cells have a higher level of PI than glioblastoma cancer stem cells [78]. Although it is not associated with patient prognosis in this tumor, the precise role of PI phosphorylation changes in glioblastoma tumors requires further investigation.

### 3.11. Biosynthesis of Phosphatidylglycerol and Cardiolipin and Glioblastoma Tumorigenesis

#### 3.11.1. Biosynthesis of Phosphatidylglycerol and Cardiolipin

In mitochondria, the first step of CL synthesis from CDP-DAG is the formation of phosphatidylglycerol phosphate (PGP) by phosphatidylglycerol phosphate synthase (PGPS/PGS1) from glycerol-3-phosphate and CDP-DAG (Figure 10) [178,179]. Subsequently, the phosphate group is removed from PGP to produce PG by the enzyme PGP phosphatase. The protein tyrosine phosphatase mitochondrial 1 (PTPMT1) serves as this enzyme [180]. In the final reaction, CL is synthesized from CDP-DAG and PG by cardiolipin synthase 1 (CLS1/CRLS1) [181,182,183]. CLS1 also has lysophosphatidylglycerol (lysoPG) acyltransferase activity but not lysoCL acyltransferase activity [184]. This enzyme processes lysoPG into LG, which is then a substrate for CL production.

#### 3.11.2. Biosynthesis of Phosphatidylglycerol and Cardiolipin in Glioblastoma

Studies on C6 rat glioma cells have shown that cardiolipin synthase and CL levels are not significant in glioma cell proliferation [185]. However, CL remodeling is significant in glioma cancer processes. Therefore, reducing the expression of tafazzin, an enzyme responsible for CL remodeling, decreases oxidative phosphorylation and reduces glioma cancer cell proliferation [185,186,187]. However, the mechanism by which tafazzin affects proliferation is unclear. Analysis using the GEPIA portal shows that CLS1 and PTPMT1 expression is higher in glioblastoma tumor tissue than in healthy brain tissue (Table 11) [48]. On the other hand, PGPS/PGS1 expression does not change. Similar results were obtained in transcriptomic analysis by Seifert et al. [49]. However, according to GEPIA, the level of expression of enzymes involved in CL biosynthesis is not related to the prognosis of glioblastoma patients [48]. This indicates that CL production is increased in glioblastoma compared to healthy brain tissue.

## 4. Synthesis of Triacylglycerol and Glioblastoma Tumorigenesis

### 4.1. Diacylglycerol O-acyltransferases and Glioblastoma Tumorigenesis

#### 4.1.1. Diacylglycerol O-acyltransferases

DAG is not only a substrate for the production of glycerophospholipids but also for TAG. Diacylglycerol O-acyltransferases (DGAT) are the enzymes responsible for this process. In humans and other mammals, there are two such enzymes: DGAT1 [188] and DGAT2 [189]. Both enzymes catalyze the same reaction: the transfer of an acyl residue to DAG. They can also participate in the synthesis of monoalkyl-diacylglycerol, thus contributing to ether lipid biosynthesis [190]. These enzymes compensate for each other in case of insufficient expression of either one [191]. However, the gene and protein sequences of DGAT1 and DGAT2 show low similarity to each other [192]. These enzymes have the same activity through functional convergence, but they belong to different gene families.

#### 4.1.2. Diacylglycerol O-acyltransferases in Glioblastoma

In the glioblastoma tumor, the expression of DGAT1 is higher compared to healthy brain tissue (Table 12) [135]. However, other sources suggest that the expression of DGAT1 does not change [48,49]. The expression of DGAT1 in the glioblastoma tumor is much higher than that of DGAT2 [135]. According to Seifert et al., the expression of DGAT2 in the glioblastoma tumor may decrease compared to healthy brain tissue [49], although this finding is not confirmed by GEPIA [48]. These results suggest that DGAT1 is the main enzyme involved in TAG biosynthesis in glioblastoma. Furthermore, higher expression of DGAT1 in glioblastoma may be associated with worse prognosis for glioblastoma patients [135], although GEPIA does not associate the expression of DGAT1 or DGAT2 with prognosis in glioblastoma patients [48].

Cancer nonstem cells in glioblastoma have more TAG than cancer stem cells, indicating where the described pathway plays a protumorigenic role [78]. DGAT1 efficiently utilizes synthesized fatty acids, thereby protecting glioblastoma cancer cells from lipotoxicity caused by high levels of fatty acids. TAG also has a protumorigenic role, as it builds lipid droplets and serves as an energy reservoir [154]. Lipid droplets also exhibit chemoresistance properties by accumulating lipophilic anticancer drugs [193]. Consequently, these drugs are not available in other parts of glioblastoma cancer cells, where they could exert their antitumor effects. A more detailed description of the role of lipid droplets in glioblastoma is provided in the section dedicated to lipid droplets.

### 4.2. Monoacylglycerol Acyltransferases and Glioblastoma Tumorigenesis

#### 4.2.1. Monoacylglycerol Acyltransferases

In addition to the pathway described above, another pathway for TAG synthesis is possible. 2-monoacylglycerol can undergo acylation by monoacylglycerol O-acyltransferase (MOGAT) to form 1,2-DAG [194]. This compound can then be converted to TAG by DGAT or MOGAT. These enzymes also exhibit DGAT activity towards 1,2-DAG and 1,3-DAG, catalyzing two steps in the biosynthesis of TAG from monoacylglycerol [194]. The highest activity of these enzymes is observed in the intestine, but lower activity is also present in the stomach, kidney, adipose tissue, and liver [195]. The main role of these enzymes is the absorption of fatty acids in the small intestine. There are three isoforms of MOGAT: MOGAT1 [195], MOGAT2 [196], and MOGAT3 [197].

#### 4.2.2. Monoacylglycerol Acyltransferases in Glioblastoma

The expression of MOGAT in the brain is very low [195]. Similarly, in glioblastoma tumors, the expression of MOGAT is also very low [48]. Therefore, it can be assumed that MOGAT plays no role in the tumorigenic processes in glioblastoma.

### 4.3. Lipid Droplets and Triacylglycerol in Glioblastoma

Lipid droplets are cellular organelles composed mainly of TAG and cholesterol esters, surrounded by a layer of phospholipids and proteins essential for their function [198]. In glioblastoma, there are significantly more lipid droplets than in low-grade glioma and healthy brain tissue [134,199]. The increased number of lipid droplets in glioblastoma is associated with higher expression of enzymes responsible for the synthesis of their components, including sterol O-acyltransferase 1 (SOAT1), which is responsible for producing cholesterol esters [134] and DGAT1, which is responsible for producing TAG [135]. However, lipid droplets are present in glioblastoma cancer nonstem cells [78], while glioblastoma cancer stem cells have fewer lipid droplets.

Cells with lipid droplets are located in the glioblastoma around necrotic regions [78,200,201], which is associated with hypoxia [78,199,201] and nutrient deprivation [202], leading to autophagy and the accumulation of lipids in lipid droplets [203]. In particular, under hypoxia, glioblastoma cancer cells take up extracellular vesicles from the tumor microenvironment, which is a source of lipids for the formation of lipid droplets [204]. Lipid droplets serve as an energy reservoir that is utilized during nutrient deprivation. During glucose deprivation, CHKα2 is phosphorylated at Ser^279^ by AMP-activated protein kinase (AMPK) and acetylated at Lys^247^ by lysine acetyltransferase 5 (KAT5) [154]. As a result, CHKα2 phosphorylates PLIN2 at Tyr^232^ and PLIN3 at Tyr^251^, directing these proteins toward proteolytic degradation. These proteins are associated with lipid droplets, and their degradation leads to lipid droplet lipolysis [154]. The released fatty acids are then subject to β-oxidation to obtain energy for glioblastoma cancer cells during nutrient deprivation [202,205].

Lipid droplets also serve as a mechanism that protects glioblastoma cancer cells from lipotoxicity resulting from excessive production and levels of free fatty acids [135,206]. In glioblastoma tumors, there is high expression of FASN and intensive production of fatty acids [12,13,42,43], which, in their free form or incorporated into phospholipids in cell membranes, can have a destructive effect on the cell’s lipid membrane structure. To counteract this, excess fatty acids are incorporated into TAG and stored in lipid droplets [135,206,207], and these fatty acids are subsequently released from these organelles and undergo β-oxidation, which drives glioblastoma cancer cell proliferation [207].

Lipid droplets are critical for glioblastoma, as evidenced by the association of patient outcomes with the number of these organelles present in the tumor. Greater numbers of lipid droplets in glioblastoma tumors are associated with poorer patient outcomes [134]. This is partly due to the fact that lipid droplets can act as a mechanism of chemoresistance. They provide a lipophilic environment within glioblastoma cancer cells, which causes poorly water-soluble chemotherapeutic drugs to accumulate in lipid droplets rather than other parts of the cell. Therefore, these drugs will have significantly weakened effects, as demonstrated in experiments using curcumin [193]. This is problematic, as the glioblastoma tumor is protected by the blood-brain barrier (BBB), particularly against drugs that are well-soluble in water [208]. The BBB is permeable to small lipophilic compounds; however, after penetrating the BBB, such compounds are accumulated in lipid droplets and thus may not act on other elements of glioblastoma cancer cells.

## 5. Conclusions and Perspective for Future Research

The biosynthesis of fatty acids, glycerophospholipids, and TAG has been extensively studied in physiological models. However, it remains poorly understood in glioblastoma and glioblastoma cancer cells. Therefore, in many parts of this article, we had to rely on experimental studies on other cancer and physiological models. We then used the GEPIA portal, which is based on TCGA data, to analyze and interpret the findings. The results demonstrate that there is still much to be discovered in the lipid metabolism of glioblastoma, with many enzymes displaying altered expression levels that are associated with poorer prognoses, according to GEPIA. However, the precise underlying mechanisms remain largely unknown, presenting a significant area for future research in the coming years.

## Figures and Tables

**Figure 1 cancers-15-02183-f001:**
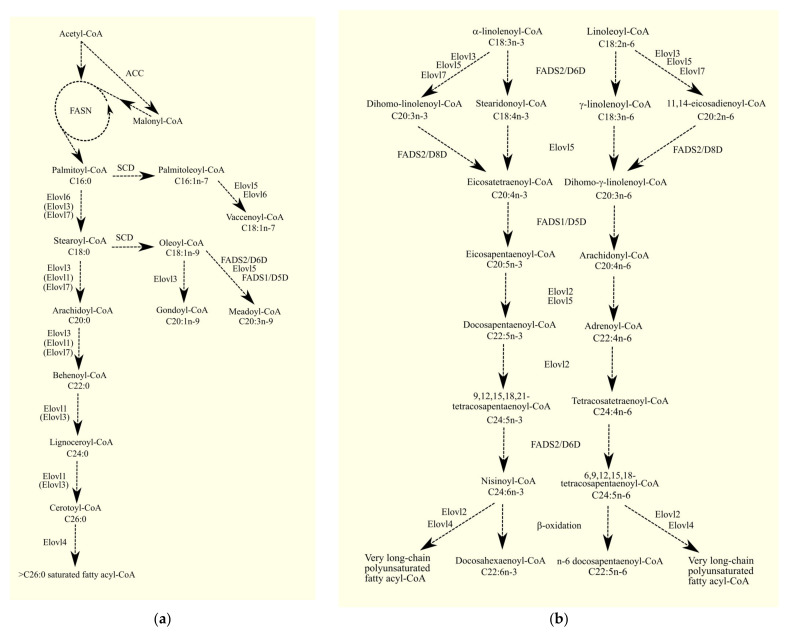
Pathways of fatty acid synthesis. (**a**) The majority of fatty acids are synthesized de novo. Initially, FASN synthesizes palmitoyl-CoA C16:0, but it is unable to elongate the chain further. Therefore, elongases are responsible for the biosynthesis of SFAs longer than 16 carbons. These enzymes elongate fatty acyl-CoA chains by two carbons. Additionally, double bonds can be introduced into acyl-CoA by desaturases. SCD is responsible for the conversion of SFAs to MUFAs, producing oleoyl-CoA C18:1n-9 from stearoyl-CoA C18:0 and palmitoleoyl C16:1n-7 from palmitoyl-CoA C16:0. The desaturases FADS1/D5D and FADS2/D6D are involved in the formation of PUFA, such as meadyl-CoA C20:3n-9, from oleoyl-CoA C18:1n-9. (**b**) In the case of omega-3 and omega-6 PUFA, these fatty acids are formed from other PUFAs of the same series. The elongation of PUFA chains involves Elovl2 and Elovl5 elongases and FADS1/D5D and FADS2/D6D desaturases. DHA is formed through β-oxidation.

**Figure 2 cancers-15-02183-f002:**
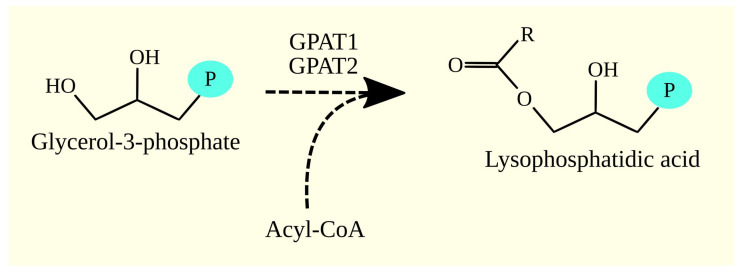
GPAT enzymes participate in the biosynthesis of lysophosphatidic acid from glycerol-3-phosphate. They require fatty acyl-CoA to catalyze this reaction.

**Figure 3 cancers-15-02183-f003:**
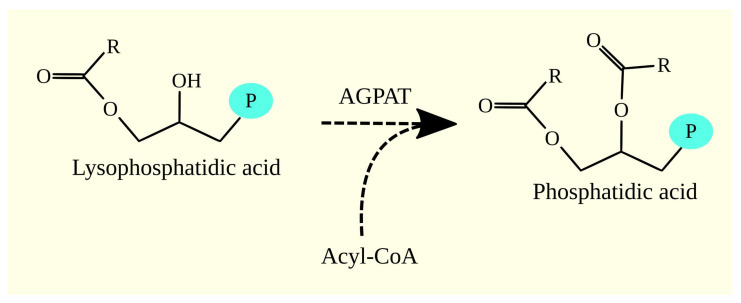
AGPAT enzymes participate in the biosynthesis of phosphatidic acid from lysophosphatidic acid. They require fatty acyl-CoA to catalyze this reaction.

**Figure 4 cancers-15-02183-f004:**
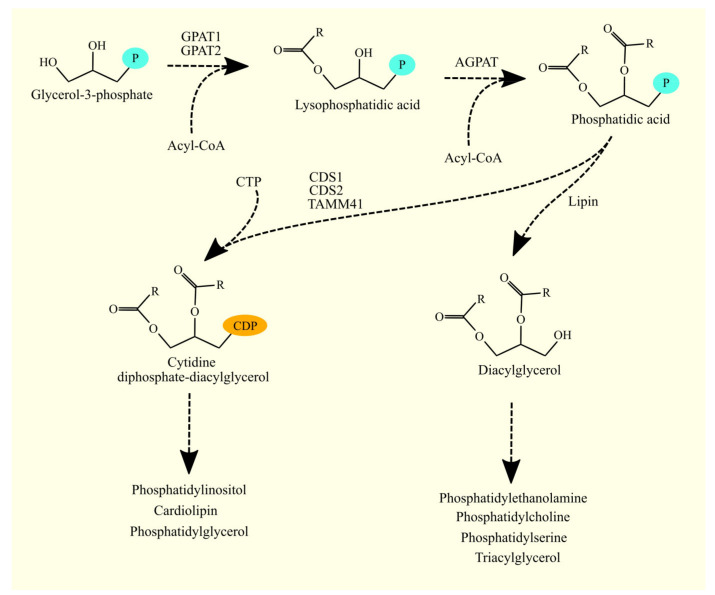
Glycerophospholipid and TAG synthesis. The synthesis of these compounds begins with glycerol-3-phosphate. The acyl group is transferred to the *sn*-1 position of this compound by GPAT, resulting in lysophosphatidic acid (1-acylglycerol-3-phosphate). In the next reaction, the second acyl group from an acyl-CoA is transferred to the *sn*-2 position by AGPAT, producing phosphatidic acid. Phosphatidic acid can undergo two different reactions. Its phosphate group can be removed by lipin to generate DAG, which can then produce PE, PC, PS, and TAG. Alternatively, phosphatidic acid can be converted to CDP-DAG by CDS1, CDS2, or in the mitochondria by TAMM41. CDP-DAG can then be converted to PI or, in the mitochondria, to PG or CL.

**Figure 5 cancers-15-02183-f005:**
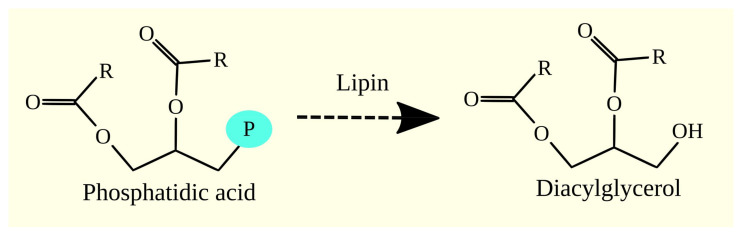
Lipins participate in the de novo synthesis of glycerophospholipids and triacylglycerol during the generation of diacylglycerol. They catalyze the reaction converting phosphatidic acid to diacylglycerol.

**Figure 6 cancers-15-02183-f006:**
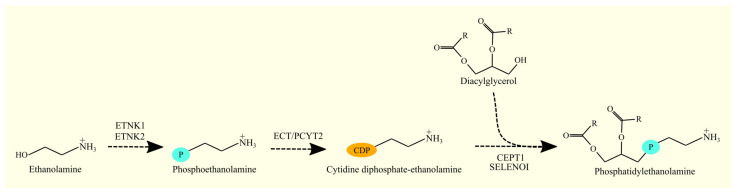
The biosynthesis of PE in the Kennedy pathway. In this pathway, ethanolamine is phosphorylated and then converted to CDP-ethanolamine. In the final step, CDP-ethanolamine reacts with DAG to produce PE. The enzymes responsible for these reactions are CEPT1 and SELENOI.

**Figure 7 cancers-15-02183-f007:**
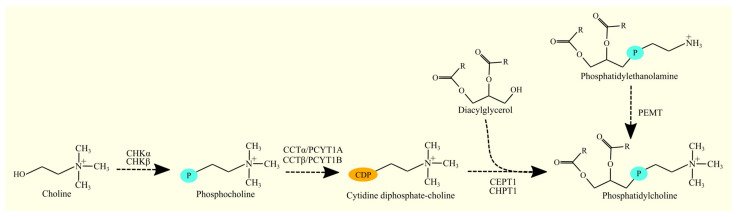
Biosynthesis of PC. The biosynthesis of PC occurs in the Kennedy pathway. In this pathway, choline is phosphorylated and then converted to CDP-choline. In the final step, CDP-choline reacts with DAG to produce PC. The enzymes responsible for these reactions are CEPT1 and CHPT1. PE and PC can undergo further transformations. Ethanolamine in PE can be methylated by PEMT to form PC from PE.

**Figure 8 cancers-15-02183-f008:**
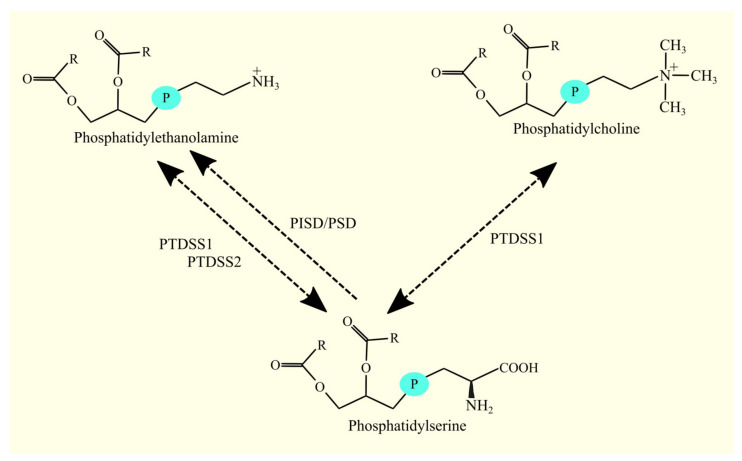
Biosynthesis of PS. Ethanolamine and choline in PE and PC can be exchanged for serine by PTDSS1 and PTDSS2, resulting in the formation of PS. PS can also undergo decarboxylation by PISD/PSD, converting this glycerophospholipid into PE.

**Figure 9 cancers-15-02183-f009:**
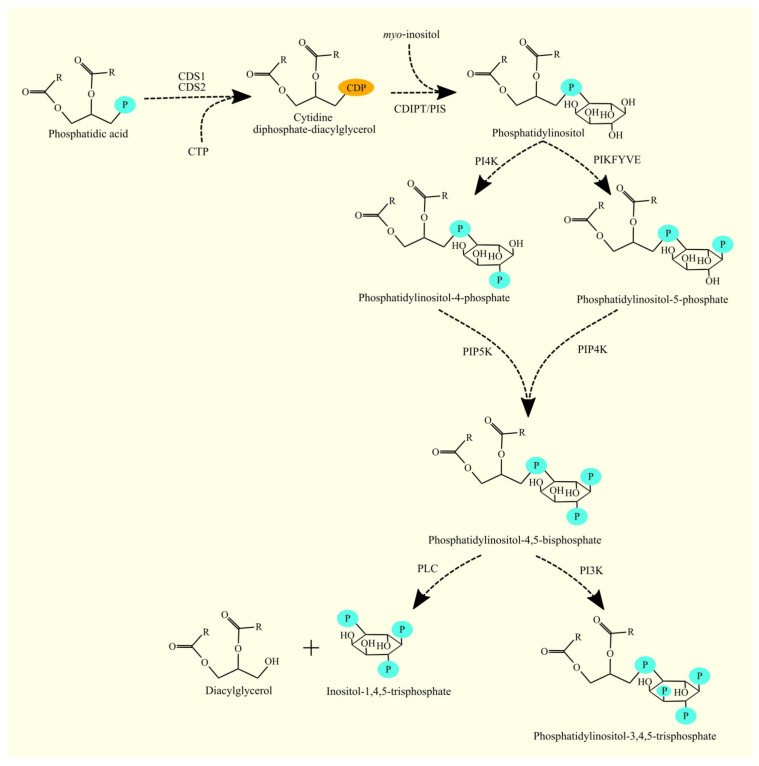
Synthesis of PI. Phosphatidic acid is converted to CDP-DAG. In the endoplasmic reticulum, this reaction is catalyzed by CDS1 and CDS2. In the endoplasmic reticulum, CDP-DAG is converted to PI by CDIPT/PIS, after which this glycerophospholipid is phosphorylated on the inositol ring by PIKFYVE and PI4K, resulting in the formation of phosphatidylinositol-5-phosphate and phosphatidylinositol-4-phosphate, respectively. These glycerophospholipids are then further phosphorylated by PIP4K and PIP5K, producing PIP_2_, which plays a role in intracellular signal transduction. It can be further phosphorylated by phosphatidylinositol-4,5-bisphosphate 3-kinase (PI3K) to phosphatidylinositol-3,4,5-trisphosphate (PIP_3_). PIP_2_ can also be converted to DAG and inositol-1,4,5-trisphosphate (IP_3_) by phospholipase C (PLC).

**Figure 10 cancers-15-02183-f010:**
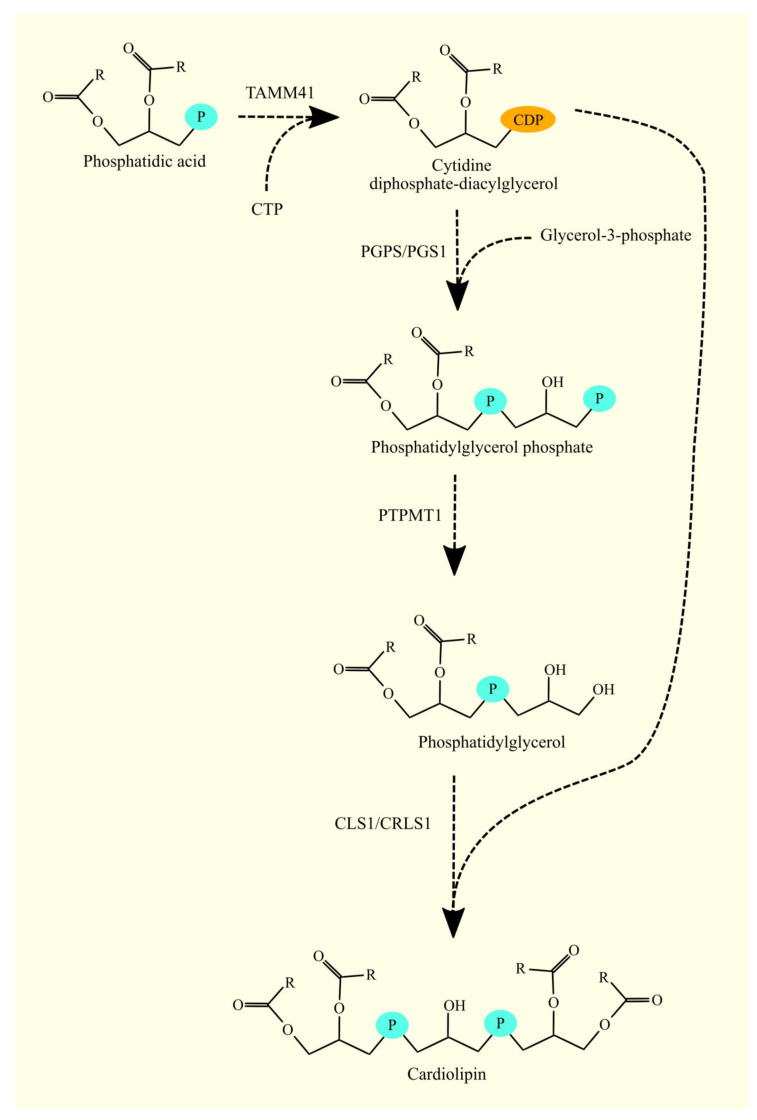
Synthesis of CL. Phosphatidic acid is converted to CDP-DAG. In mitochondria, this reaction is catalyzed by TAMM41. In mitochondria, PG and CL are produced from CDP-DAG. In the first step, PGP is produced from CDP-DAG and glycerol-3-phosphate by PGPS/PGS1. The phosphate group is then removed from this compound by PTPMT1, producing PG. CL is then synthesized from PG and CDP-DAG with the participation of CLS1/CRLS1.

**Table 1 cancers-15-02183-t001:** Characteristics of FASN and ACC.

Enzyme	Properties	Expression Level in Glioblastoma Tumor Relative to Healthy Brain Tissue			Impact on Survival Rate	Comments
Source		GEPIA [48]	Seifert et al. [49]	Other Data Source	GEPIA [48]	1.
FASN	Synthesis of acyl-CoA to a length of 16 carbons	Expression does not change	Expression does not change	Higher expression in the tumor [12,13,42,43]	No significant impact on prognosis	Expression higher by *IDH1* mutation, hypoxia reduces expression, and higher expression in glioblastoma cancer stem cells
ACC	Production of malonyl-CoA, a substrate for FASN and elongase	Expression does not change	Lower expression in the tumor		No significant impact on prognosis	Hypoxia reduces expression

Blue background—expression in the tumor lower than in healthy brain tissue; red background—expression in the tumor higher than in healthy brain tissue; red background—higher expression in the tumor means a worse prognosis.

**Table 2 cancers-15-02183-t002:** Characteristics of elongases involved in the biosynthesis of fatty acids.

Enzyme	Properties	Expression Level in Glioblastoma Tumor Relative to Healthy Brain Tissue			Impact on Survival Rate	Comments
Source		GEPIA [48]	Seifert et al. [49]	Other Data Source	GEPIA [48]	2.
Elovl1	Elongation of saturated acyl-CoA	Higher expression in the tumor	Expression does not change	Lower expression in the tumor [52]	Worse prognosis	Hypoxia reduces expression
Elovl2	Elongation of 20- and 22-carbon polyunsaturated acyl-CoA	Higher expression in the tumor	Expression does not change	Higher expression in the tumor [54];	No significant impact on prognosis	Higher expression in glioblastoma cancer stem cells
				Expression does not change [52]		
Elovl3	Elongation of saturated acyl-CoA	Expression does not change	Expression does not change	Expression does not change [52]	Worse prognosis	Hypoxia reduces expression
Elovl4	Elongation of very long-chain fatty acyl-CoA	Expression does not change	Expression does not change	Expression does not change [52]	No significant impact on prognosis	
Elovl5	Elongation of 18- and 2-carbon polyunsaturated acyl-CoA	Higher expression in the tumor	Higher expression in the tumor	Expression does not change [52]	No significant impact on prognosis	
Elovl6	Elongation of palmitoyl-CoA C16:0	Expression does not change	Expression does not change	Expression does not change [52]	No significant impact on prognosis	
Elovl7	Elongation of saturated acyl-CoA	Lower expression in the tumor	Lower expression in the tumor	Lower expression in the tumor [52]	No significant impact on prognosis	Hypoxia reduces expression

Blue background—expression in the tumor lower than in healthy brain tissue; red background—expression in the tumor higher than in healthy brain tissue; red background—higher expression in the tumor means a worse prognosis.

**Table 3 cancers-15-02183-t003:** Characteristics of desaturases involved in the biosynthesis of fatty acids.

Enzyme	Properties	Expression Level in the Glioblastoma Tumor Relative to Healthy Brain Tissue			Impact on Survival Rate	Comments
Source		GEPIA [48]	Seifert et al. [49]	Other Data Source	GEPIA [48]	3.
SCD	Desaturation of saturated acyl-CoA, MUFA formation	Expression does not change	Lower expression in the tumor	Lower expression in the tumor [68,69]	No significant impact on prognosis	Hypoxia increases expression; Higher expression in *IDH1* mutation
SCD5	Desaturation of saturated acyl-CoA, formation of MUFAs	Higher expression in the tumor	Expression does not change	Expression does not change [68]	No significant impact on prognosis	
FADS1	Insertion of a double bond into polyunsaturated acyl-CoA	Expression does not change	Expression does not change	Expression does not change [68]	No significant impact on prognosis	Higher expression in glioblastoma cancer stem cells
FADS2	Insertion of a double bond into polyunsaturated acyl-CoA	Higher expression in the tumor	Higher expression in the tumor	Lower expression in the tumor [68]	No significant impact on prognosis	Higher expression in glioblastoma cancer stem cells
FADS3	Little known	Expression does not change	Expression does not change	Expression does not change [68]	Worse prognosis	

Blue background—expression in the tumor lower than in healthy brain tissue; red background—expression in the tumor higher than in healthy brain tissue; red background—higher expression in the tumor means a worse prognosis.

**Table 4 cancers-15-02183-t004:** Characteristics of GPAT, enzymes involved in the biosynthesis of lysophosphatidic acid from glycerol-3-phosphate.

Enzyme	Properties	Expression Level in the Glioblastoma Tumor Relative to Healthy Brain Tissue		Impact on Survival Rate
Source		GEPIA [48]	Seifert et al. [49]	GEPIA [48]
GPAT1	Mitochondrial enzyme	Expression does not change	Expression does not change	No significant impact on prognosis
GPAT2	Mitochondrial enzyme	Expression does not change		No significant impact on prognosis
GPAT3	Enzyme in endoplasmic reticulum, also 1-acylglycerol-3-phosphate O-acyltransferase activity; Questionable GPAT activity; Other name AGPAT10 and AGPAT9	Expression does not change		Worse prognosis
GPAT4	An enzyme in the endoplasmic reticulum, also 1-acylglycerol-3-phosphate O-acyltransferase activity; Other name AGPAT6	Higher expression in the tumor	Expression does not change	No significant impact on prognosis

Red background—expression in the tumor higher than in healthy brain tissue; red background—higher expression in the tumor means a worse prognosis.

**Table 5 cancers-15-02183-t005:** Characteristics of AGPAT, enzymes involved in the biosynthesis of phosphatidic acid from lysophosphatidic acid.

Enzyme	Properties	Expression Level in the Glioblastoma Tumor Relative to Healthy Brain Tissue		Impact on Survival Rate
Source		GEPIA [48]	Seifert et al. [49]	GEPIA [48]
AGPAT1	Localization in the endoplasmic reticulum	Expression does not change	Expression does not change	No significant impact on prognosis
AGPAT2	Localization in the endoplasmic reticulum	Expression does not change	Expression does not change	No significant impact on prognosis
AGPAT3	Localization in the endoplasmic reticulum, lysophospholipids acyltransferase activity	Expression does not change	Lower expression in the tumor	No significant impact on prognosis
AGPAT4	Localization in the endoplasmic reticulum	Expression does not change	Lower expression in the tumor	No significant impact on prognosis
AGPAT5	Localization in the endoplasmic reticulum and mitochondria, lysophospholipids acyltransferase activity	Higher expression in the tumor	Higher expression in the tumor	No significant impact on prognosis
AGPAT6	Localization in the endoplasmic reticulum and on lipid droplets; GPAT activity, also called GPAT4	Higher expression in the tumor	Expression does not change	No significant impact on prognosis
AGPAT7	Localization in the endoplasmic reticulum, Introduces DHA C22:6n-3 into lysophospholipids; Other name LPEAT2 and LPCAT4	Lower expression in the tumor	Lower expression in the tumor	No significant impact on prognosis
AGPAT8	Localization in the endoplasmic reticulum, acyl-CoA:lysocardiolipin acyltransferase activity; Other name ALCAT1 and LCLAT1	Higher expression in the tumor	Expression does not change	No significant impact on prognosis
AGPAT9	Localization in the endoplasmic reticulum and on lipid droplets, lysophospholipids acyltransferase activity, production of dipalmitoylphosphatidylcholine; Other name LPCAT1	Higher expression in the tumor	Higher expression in the tumor	No significant impact on prognosis
AGPAT10	Localization in the endoplasmic reticulum, GPAT activity, also called GPAT3, AGPAT9	Expression does not change		Worse prognosis
AGPAT11	Localization in the endoplasmic reticulum and on lipid droplets, lysophospholipids acyltransferase activity another name for LPCAT2	Higher expression in the tumor	Expression does not change	No significant impact on prognosis

Blue background—expression in the tumor lower than in healthy brain tissue; red background—expression in the tumor higher than in healthy brain tissue; red background—higher expression in the tumor means a worse prognosis.

**Table 6 cancers-15-02183-t006:** Characteristics of lipins, enzymes involved in DAG biosynthesis from phosphatidic acid.

Enzyme	Properties	Expression Level in the Glioblastoma Tumor Relative to Healthy Brain Tissue		Impact on Survival Rate
Source		GEPIA [48]	Seifert et al. [49]	GEPIA [48]
Lipin 1	Expression increased by hypoxia [115] It interacts with about 30 proteins, including PPARα and PPARγ [116]	Expression does not change	Lower expression in the tumor	No significant impact on prognosis
Lipin 2	Reduction in NLRP3 activation, decrease in P2X7 activation [117]	Expression does not change	Expression does not change	No significant impact on prognosis
Lipin 3		Expression does not change	Expression does not change	No significant impact on prognosis

Blue background—expression in the tumor lower than in healthy brain tissue.

**Table 7 cancers-15-02183-t007:** Characteristics of enzymes involved in PE biosynthesis.

Enzyme	Properties	Expression Level in the Glioblastoma Tumor Relative to Healthy Brain Tissue		Impact on Survival Rate
Source		GEPIA [48]	Seifert et al. [49]	GEPIA [48]
ETNK1	Production of phosphoethanolamine	Expression does not change	Expression does not change	Better prognosis
ETNK2	Generation of phosphoethanolamine, slight choline kinase activity, expression reduced by *IDH1* mutation	Expression does not change	Expression does not change	Worse prognosis
ECT/PCYT2	Production of CDP-ethanolamine	Expression does not change	Lower expression in the tumor	No significant impact on prognosis
CEPT1	Production of PE and PC in Kennedy pathway	Higher expression in the tumor	Expression does not change	Better prognosis (*p* = 0.062)
SELENOI	Production of PE and plasmanyl-PE in the Kennedy pathway	Expression does not change	Lower expression in the tumor	No significant impact on prognosis
PISD/PSD	Production of PE from PS	Expression does not change	Expression does not change	No significant impact on prognosis

Blue background—expression in the tumor lower than in healthy brain tissue; red background—expression in the tumor higher than in healthy brain tissue; blue background—higher expression in the tumor means a better prognosis; red background—higher expression in the tumor means a worse prognosis.

**Table 8 cancers-15-02183-t008:** Characteristics of enzymes involved in PC biosynthesis.

Enzyme	Properties	Expression Level in the Glioblastoma Tumor Relative to Healthy Brain Tissue		Impact on Survival Rate
Source		GEPIA [48]	Seifert et al. [49]	GEPIA [48]
CHKα	Production of phosphocholine, pro-oncogenic properties; Androgen receptor chaperone [149]; Protein kinase activity	Lower expression in the tumor	Expression does not change	No significant impact on prognosis
CHKβ	Production of phosphocholine	Expression does not change	Expression does not change	No significant impact on prognosis
CCTα/PCYT1A	Generation of CDP-choline, localization in the endoplasmic reticulum and in the cell nucleus	Expression does not change	Expression does not change	No significant impact on prognosis
CCTβ/PCYT1B	Generation of CDP-choline, localization in the endoplasmic reticulum	Expression does not change	Expression does not change	No significant impact on prognosis
CEPT1	Production of PC and PE in Kennedy pathway	Higher expression in the tumor	Expression does not change	Better prognosis (*p* = 0.062)
CHPT1	Production of PC in Kennedy pathway	Expression does not change	Higher expression in the tumor	No significant impact on prognosis
PEMT	Production PC from PE	Higher expression in the tumor	Higher expression in the tumor	No significant impact on prognosis

Blue background—expression in the tumor lower than in healthy brain tissue; red background—expression in the tumor higher than in healthy brain tissue; blue background—higher expression in the tumor means a better prognosis.

**Table 9 cancers-15-02183-t009:** Characteristics of enzymes involved in PS biosynthesis.

Enzyme	Properties	Expression Level in the Glioblastoma Tumor Relative to Healthy Brain Tissue		Impact on Survival Rate
Source		GEPIA [48]	Seifert et al. [49]	GEPIA [48]
PTDSS1	Replacing PC choline with serine, activity is not reduced by PS	Higher expression in the tumor	Lower expression in the tumor	No significant impact on prognosis
PTDSS2	Replacing choline in PC and ethanolamine in PE with serine, activity is reduced by PS	Expression does not change	Higher expression in the tumor	Worse prognosis

Blue background—expression in the tumor lower than in healthy brain tissue; red background—expression in the tumor higher than in healthy brain tissue; red background—higher expression in the tumor means a worse prognosis.

**Table 10 cancers-15-02183-t010:** Characteristics of enzyme CDP-DAG synthases.

Enzyme	Properties	Expression Level in the Glioblastoma Tumor Relative to Healthy Brain Tissue		Impact on Survival Rate
Source		GEPIA [48]	Seifert et al. [49]	GEPIA [48]
CDS1	An enzyme in the endoplasmic reticulum, PI biosynthesis pathway	Lower expression in the tumor	Lower expression in the tumor	No significant impact on prognosis
CDS2	An enzyme in the endoplasmic reticulum, PI biosynthesis pathway	Expression does not change	Expression does not change	Better prognosis
TAMM41	Enzyme in the mitochondrion, CL and PG biosynthesis pathway	Expression does not change	Expression does not change	No significant impact on prognosis

Blue background—expression in the tumor lower than in healthy brain tissue; blue background—higher expression in the tumor means a better prognosis.

**Table 11 cancers-15-02183-t011:** Characteristics of enzymes involved in the biosynthesis of CL and PG.

Enzyme	Properties	Expression Level in the Glioblastoma Tumor Relative to Healthy Brain Tissue		Impact on Survival Rate
Source		GEPIA [48]	Seifert et al. [49]	GEPIA [48]
PGPS/PGS1	Biosynthesis of phosphatidylglycerol phosphate from CDP-DAG and glycerol-3-phosphate	Expression does not change	Expression does not change	No significant impact on prognosis
PTPMT1	Generation of phosphatidylglycerol from phosphatidylglycerol phosphate	Higher expression in the tumor	Higher expression in the tumor	No significant impact on prognosis
CLS1/CRLS1	Biosynthesis of CL from phosphatidylglycerol and CDP-DAG lysophosphatidylglycerol acyltransferase activity	Higher expression in the tumor	Higher expression in the tumor	No significant impact on prognosis

Red background—expression in the tumor higher than in healthy brain tissue.

**Table 12 cancers-15-02183-t012:** Characteristics of enzymes involved in TAG biosynthesis.

Enzyme	Properties	Expression Level in the Glioblastoma Tumor Relative to Healthy Brain Tissue			Impact on Survival Rate
Source		GEPIA [48]	Seifert et al. [49]	Cheng et al. [135]	
DGAT1	TAG biosynthesis from DAG starvation-induced lipid droplets formation	Expression does not change	Expression does not change	Higher expression in the tumor	Worse prognosis [135]
					No significant impact on prognosis [48]
DGAT2	TAG biosynthesis from DAG lipid droplets formation	Expression does not change	Lower expression in the tumor		No significant impact on prognosis [48]

Blue background—expression in the tumor lower than in healthy brain tissue; Red background—expression in the tumor higher than in healthy brain tissue; red background—higher expression in the tumor means a worse prognosis.

## Data Availability

Not applicable.

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
