# Peer review of "Biosynthesis and Significance of Fatty Acids, Glycerophospholipids, and Triacylglycerol in the Processes of Glioblastoma Tumorigenesis"

_cancers, 2023, doi:10.3390/cancers15072183_

Round 1
Reviewer 1 Report
This review is very well conducted. In my opinion, it is interesting, and I recommend its publication in the Journal of Cancers after the following points:
-Introduce the abbreviation for the first time.
- The authors present more figures and Tables for better clarification.
- Improve the resolution of the graphical abstract, also add the signaling pathway in the figure.
Author Response
This review is very well conducted. In my opinion, it is interesting, and I recommend its publication in the Journal of Cancers after the following points:
-Introduce the abbreviation for the first time.
Additional abbreviations have been added to the article
- The authors present more figures and Tables for better clarification.
Addiditional figures have been added to the article
- Improve the resolution of the graphical abstract, also add the signaling pathway in the figure.
The low quality of the graphical abstract is due to the PDF formatting issues. The size of the graphical abstract has been increased which should improve the quality of the image.

Reviewer 2 Report
A well presented paper, focusing on a major subject concerning health, and analyzing a special methabolyte class.
Author Response
Thank you very much for review
Reviewer 3 Report
Manuscript Title: Biosynthesis and significance of fatty acids, glycerophospholipids, and triacylglycerol in the processes of glioblastoma tumorigenesis
. . . . . . . . . . . . . . . . . . . . . . . . . . . . . . . . . . . . . . . . . . . . . . . . . . . . . . . . . . . . . . . . . . . . .
Comments on the current review article on the biosynthesis of fatty acids, glycerophospholipids, and triacylglycerol and their importance in the processes of glioblastoma tumorigenesis are presented below.
After considering the biosynthesis of fatty acids, glycerophospholipids and triacylglycerol as a review in itself, it seems to have increased the current load of the article to mention its importance in glioblastoma tumorigenesis processes. Because when reading the biosynthesis section of the relevant articles, it creates the opinion that a different subject than Glioblastoma has been entered in depth.
Authors are advised to handle the article in a manner that does not tire the reader.
In line 25, “We also also describe the significance of individual” statement should be revised in view of grammatical rules.
In the last paragraph of the introduction, no references to the necessity of the relationship between glioblastoma and fatty acids are presented.
Since this review article is so long and heavy, it seems difficult for the reader to follow the topic integrity.
There is a high rate of reference usage.
. . . . . . . . . . . . . . . . . . . . . . . . . . . . . . . . . . . . . . . . . . . . . . . . . . . . . . . . . . . . . . . . . . . . .
Author Response
Comments on the current review article on the biosynthesis of fatty acids, glycerophospholipids, and triacylglycerol and their importance in the processes of glioblastoma tumorigenesis are presented below.
After considering the biosynthesis of fatty acids, glycerophospholipids and triacylglycerol as a review in itself, it seems to have increased the current load of the article to mention its importance in glioblastoma tumorigenesis processes. Because when reading the biosynthesis section of the relevant articles, it creates the opinion that a different subject than Glioblastoma has been entered in depth.
The significance of the synthesis of fatty acids, glycerophospholipids, and triacylglycerol in glioblastoma cancer processes has been poorly understood. To better acquaint the reader with this issue, we placed greater emphasis than usual on the theoretical introductions to each of the subsections. In these introductions, we described the physiological significance of individual enzymes responsible for the synthesis of fatty acids, glycerophospholipids, and triacylglycerol. We could not include this information in the subsections on glioblastoma because the described properties were not investigated in the glioblastoma model. We then suggested that the described enzymes may also have the same functions in glioblastoma. This writing style was explained in the section on the methods used.
Authors are advised to handle the article in a manner that does not tire the reader.
In line 25, “We also also describe the significance of individual” statement should be revised in view of grammatical rules.
The sentence has been corrected. The entire article has been rechecked by the translator.
In the last paragraph of the introduction, no references to the necessity of the relationship between glioblastoma and fatty acids are presented.
The article has been corrected according to the reviewer's recommendation.
Since this review article is so long and heavy, it seems difficult for the reader to follow the topic integrity.
A table of contents has been added to the article to help readers better navigate and locate specific sections within the article.
There is a high rate of reference usage.
The paper discusses a very broad topic, which is the metabolism of the most important lipids. To cover this entire subject, it was necessary to cite a large number of experimental articles.

Round 2
Reviewer 3 Report
Manuscript Title: Biosynthesis and significance of fatty acids, glycerophospholipids, and triacylglycerol in the processes of glioblastoma tumorigenesis
Review R1
. . . . . . . . . . . . . . . . . . . . . . . . . . . . . . . . . . . . . . . . . . . . . . . . . . . . . . . . . . . . . . . . . . . . . . . . . . . . . . . . . . . . . .
The following are comments on the recent review article on the biosynthesis of fatty acids, glycerophospholipids, and triacylglycerol and their importance in the processes of glioblastoma tumorigenesis.
This review article is so long and comprehensive that it is difficult for the reader to follow it completely. The reasons for this are the following.
After discussing the biosynthesis of fatty acids, glycerophospholipids and triacylglycerol alone, mentioning their importance in glioblastoma tumorigenesis seems to increase the length of the article.
This is because when reading the biosynthesis section of the article, one gets the feeling that a topic other than glioblastoma is being covered in depth.
The authors are advised to structure the article in a way that does not tire the reader. And there are still a lot of references.
. . . . . . . . . . . . . . . . . . . . . . . . . . . . . . . . . . . . . . . . . . . . . . . . . . . . . . . . . . . . . . . . . . . . . . . . . . . . . . . . . . . . . .
Author Response
The following are comments on the recent review article on the biosynthesis of fatty acids, glycerophospholipids, and triacylglycerol and their importance in the processes of glioblastoma tumorigenesis.
This review article is so long and comprehensive that it is difficult for the reader to follow it completely. The reasons for this are the following.
After discussing the biosynthesis of fatty acids, glycerophospholipids and triacylglycerol alone, mentioning their importance in glioblastoma tumorigenesis seems to increase the length of the article.
This is because when reading the biosynthesis section of the article, one gets the feeling that a topic other than glioblastoma is being covered in depth.
The authors are advised to structure the article in a way that does not tire the reader. And there are still a lot of references.
Thank you very much for your comments, as suggested by the reviewer, the manuscript was shortened by 7 pages and 130 bibliographic items.

Round 3
Reviewer 3 Report
Manuscript Title: Biosynthesis and significance of fatty acids, glycerophospholipids, and triacylglycerol in the processes of glioblastoma tumorigenesis
Review R3
. . . . . . . . . . . . . . . . . . . . . . . . . . . . . . . . . . . . . . . . . . . . . . . . . . . . . . . . . . . . . . . . . . . . .
Below are comments on the recent review article on the biosynthesis of fatty acids, glycerophospholipids, and triacylglycerol and their importance in glioblastoma tumorigenesis processes.
It was observed that the authors made the necessary corrections in line with the comments. It was also deemed appropriate by the authors to add a table of contents to the article.
. . . . . . . . . . . . . . . . . . . . . . . . . . . . . . . . . . . . . . . . . . . . . . . . . . . . . . . . . . . . . . . . . . . . .